# Two parallel pathways connect glutamine metabolism and mTORC1 activity to regulate glutamoptosis

Clément Bodineau [1,2], Mercedes Tomé[1], Sarah Courtois [3], Ana S. H. Costa [4,5], Marco Sciacovelli [4], Benoit Rousseau[6], Elodie Richard[7], Pierre Vacher [7], Carlos Parejo-Pérez[8], Emilie Bessede[3], Christine Varon[3], Pierre Soubeyran[7], Christian Frezza [4], Piedad del Socorro Murdoch[1,9], Victor H. Villar [10] & Raúl V. Durán [1,2,7✉]

Glutamoptosis is the induction of apoptotic cell death as a consequence of the aberrant activation of glutaminolysis and mTORC1 signaling during nutritional imbalance in proliferating cells. The role of the bioenergetic sensor AMPK during glutamoptosis is not defined yet. Here, we show that AMPK reactivation blocks both the glutamine-dependent activation of mTORC1 and glutamoptosis in vitro and in vivo. We also show that glutamine is used for asparagine synthesis and the GABA shunt to produce ATP and to inhibit AMPK, independently of glutaminolysis. Overall, our results indicate that glutamine metabolism is connected with mTORC1 activation through two parallel pathways: an acute alpha-ketoglutarate-dependent pathway; and a secondary ATP/AMPK-dependent pathway. This dual metabolic connection between glutamine and mTORC1 must be considered for the future design of therapeutic strategies to prevent cell growth in diseases such as cancer.

[1] Centro Andaluz de Biología Molecular y Medicina Regenerativa—CABIMER, Consejo Superior de Investigaciones Científicas, Universidad de Sevilla, Universidad Pablo de Olavide, Seville, Spain. [2] Institut Européen de Chimie et Biologie, INSERM U1218, Université de Bordeaux, Pessac, France. [3] Bordeaux Research in Translational Oncology, INSERM U1053, Université de Bordeaux, Bordeaux cedex, France. [4] Medical Research Council Cancer Unit, Hutchison/MRC Research Centre, Box 197, Cambridge Biomedical Campus, University of Cambridge, Cambridge, UK. [5] Cold Spring Harbor Laboratory, Cold Spring Harbor, NY, USA. [6] Service Commun des Animaleries, Animalerie A2, University of Bordeaux, Bordeaux, France. [7] INSERM U1218, Institut Bergonié, Bordeaux, France. [8] Instituto de Bioquímica Vegetal y Fotosíntesis, Consejo Superior de Investigaciones Científicas, Universidad de Sevilla, Seville, Spain. [9] Departamento de Bioquímica Vegetal y Biología Molecular, Universidad de Sevilla, Seville, Spain. [10] CRUK Beatson Institute, Glasgow, UK. ✉email: raul.duran@cabimer.es

Among all amino acids, glutamine is the most abundant in the blood[1–3]. The importance of glutamine for cancer cells growth has been extensively described and linked to its role as a precursor for α-ketoglutarate (αKG) to sustain the tricarboxylic acid (TCA) cycle. Indeed, glutamine is mainly metabolized through glutaminolysis in a two-step reaction: first the enzyme glutaminase (GLS) catalyzes the hydrolysis of glutamine to glutamate; and then glutamate is converted to αKG through an oxidative deamination reaction catalyzed by the enzyme glutamate dehydrogenase (GDH). Another amino acid, leucine, acts as an allosteric activator of the second enzyme, being consequently necessary for glutaminolysis induction[4,5].

Although glutamine can participate in the synthesis of different amino acids through the production of glutamate, only asparagine requires glutamine for de novo synthesis. Asparagine synthetase (ASNS) is the enzyme that converts glutamine and aspartate into glutamate and asparagine in an ATP-dependent manner[6]. ASNS expression is upregulated during amino acid starvation through the activation of activating transcription factor 4 (ATF4)[7]. Asparagine has been identified as an exchange amino acid factor that regulates the serine/threonine kinase mammalian target of rapamycin complex 1 (mTORC1), nucleotide biosynthesis and proliferation[7].

The role of GABA shunt is a bypass for two steps of the TCA cycle, wherein GABA is synthesized from glutamate by a GAD1-encoded decarboxylase, transaminated into succinic semialdehyde, and metabolized into the TCA cycle intermediate succinate[8]. In addition to its role as a neurotransmitter, GABA has been detected in a wide range of peripheral tissues[9] and recently linked to castration-resistant prostate cancer progression through the regulation of nuclear androgen receptor by GABA[10]. That study reported that GABA shunt is upregulated in response to the activating phosphorylation of GAD65 by the PI3K pathway. In a different cancer type, the increased uptake and metabolism of GABA in breast to brain metastatic cells was linked to an increase of NADH levels in the microenvironment conferring a proliferative advantage to the tumor[11].

Previously, our results demonstrated that glutamine, and most particularly its catabolic conversion to αKG through glutaminolysis, activates mTORC1 at the short and long-term[5,12]. mTORC1 is a major signaling hub that integrates different inputs, such as nutrients, oxygen, energy, and growth factors, to regulate the metabolic pathways controlling cell growth and proliferation[13]. As mTORC1 is over-activated in 80% of solid tumors[14], different analogs of its inhibitor rapamycin have been developed to inhibit mTORC1 in patients. At a clinical level, the results remain modest and a clear need for co-therapies has been described[15–17]. Our previous work determined that the activation of mTORC1 by glutaminolysis during nutritional imbalance lead to the anomalous inhibition of autophagy and a subsequent form of apoptosis named glutamoptosis[12,18]. But still, the impact of glutamine metabolism on the bioenergetic status of the cells during long-term mTORC1 activation and glutamoptosis induction remains to be defined.

Eukaryotes have developed a system to sense low ATP levels via another serine/threonine kinase, the AMP-activated protein kinase (AMPK) complex. Under low intracellular ATP levels, AMP or ADP can directly bind to the γ regulatory subunit of AMPK, leading to a conformational change that allows the activating phosphorylation of AMPK[13]. Once activated, AMPK redirects the metabolism towards the increase of catabolism and the decrease in anabolism through phosphorylation of downstream key protein pathways, ultimately leading to mTORC1 inhibition[13].

Herein, we report that the amino acid glutamine is sufficient to sustain the production of ATP in absence of any other amino acid, following a glutaminolysis-independent mechanism. During glutamine sufficiency, ASNS and GABA shunt are responsible for metabolizing glutamine to generate ATP and to inhibit AMPK. Thus, our results indicate that glutamine activates mTORC1 following a two branches mechanism: a short-term mechanism involving αKG production, and a long-term mechanism involving ASNS, GABA, and AMPK.

## Results

**Glutaminolysis sustains the production of ATP to inhibit AMPK and to activate mTORC1.** Previously, we showed that the addition of leucine and glutamine ("LQ treatment") to amino acid-starved cells is sufficient to activate glutaminolysis and subsequently mTORC1[5,19]. To better understand the role of the bioenergetic status of the cell in this pathway, we first investigated if LQ treatment impacted the production of ATP in the cell and the activation of AMPK. Confirming our previous observations, short-term (2 h) amino acid starvation had no effect in the levels of ATP in cultured cells. However, we observed that, at longer times, amino acid withdrawal significantly decreased ATP/ADP ratio in U2OS and HCT116 cellular models (Fig. 1A, B and Supplementary 1A, B). LQ addition in amino acid-starved cells sustained the levels of ATP at 4, 8, 12, 24, 48, and 72 h (Fig. 1C and Supplementary 1C–E). By contrast, the addition of methionine or arginine, two amino acids which have been reported to activate mTORC1 at short-term, did not sustain the production of ATP (Supplementary 1F, G). Measurements of glutamine and leucine levels in the medium indicated that these amino acids remained fully available even after 72 h of LQ treatment (Supplementary 1H–I). In agreement to these previous observations, we also observed that LQ treatment increased the basal respiration of U2OS cells compared to amino acid starvation (Fig. 1D). The maximal oxygen consumption rate (OCR) was also increased in the presence of LQ, which resulted in an increase of ATP production linked to respiration (Fig. 1E). Thus, we concluded that long-term glutaminolysis, even in the absence of any other amino acid, was sufficient to increase the respiratory capacity and the production of ATP in the cell.

We next investigated if the observed production of ATP in LQ-treated cells had any impact on the phosphorylation of AMPK. The removal of amino acids from the culture medium led to the inhibition of mTORC1 in U2OS cells, as broadly reported before[20], but concomitantly it led also to the phosphorylation of AMPK at threonine 172 (Fig. 1F), in agreement with the observed decrease in ATP levels. Accordingly, the phosphorylation of AMPK was reverted by the addition of LQ, which again correlated with the increase in ATP levels in this condition (Fig. 1F). The LQ-induced inhibition of AMPK was observed at 24, 48, and 72 h (Fig. 1G). Long-term activation of mTORC1 downstream of AMPK during LQ treatment was confirmed by the phosphorylation of its target proteins S6K, S6, and 4EBP1 (Fig. 1F, G), and by its translocation at the surface of the lysosome using LAMP2 (Fig. 1H, I) or CD63 (Supplementary 1J, K) as lysosomal markers. To confirm a mechanistic role of AMPK during LQ-mediated mTORC1 activation at long-term, we investigated the status of mTORC1 by the addition of the AMPK activator AICAR in LQ-treated U2OS and HCT116 cells. As observed in Fig. 1J and Supplementary 1L, reactivation of AMPK using AICAR was sufficient to prevent the LQ-mediated activation of mTORC1. Similar results were obtained by transfecting a MYC-tagged constitutively active form of AMPK (CA-AMPK, kindly provided by Prof. Benoit Viollet, Paris, France) (Fig. 1K) or using other AMPK activators, such as metformin or A769662 (Fig. 1L). Surprisingly, the ablation of AMPK using AMPK$^{-/-}$ MEFs (again, kindly provided by Prof.

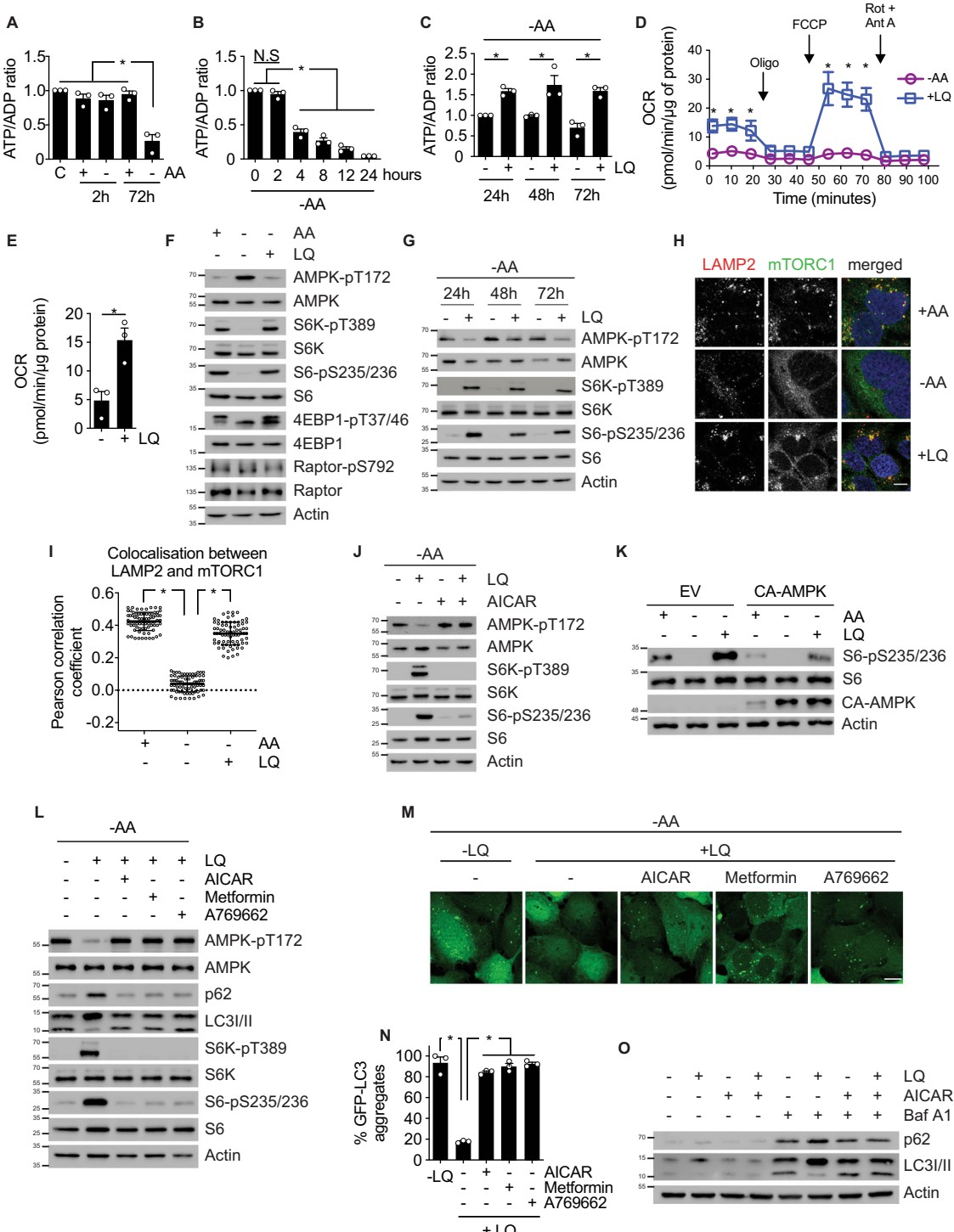

Benoit Viollet) also prevented the LQ-mediated activation of mTORC1 (Supplementary 1M). This result suggested that, in addition to acting as a negative regulator of mTORC1, the presence of AMPK is necessary for the connection between glutamine metabolism and mTORC1, underscoring the complexity of the metabolic adaptations in these circumstances.

We also confirmed that this AMPK-regulated activation of mTORC1 during long-term LQ treatment had a direct physiological impact in the cell in terms of autophagy activation, a key process during glutamoptosis induction[12]. As observed in Fig. 1M,

N, LQ-treated U2OS cells stably expressing the autophagic reporter GFP-LC3 displayed an increase of GFP-LC3 aggregates upon AMPK reactivation using AICAR, metformin, or A769662. Consistently, reactivation of AMPK using these different activators also decreased the levels of endogenous p62 and LC3I, increasing LC3II formation (Fig. 1L), well-known autophagy markers. We also evaluated the autophagic flux using bafilomycin A1 (Baf A1) to trap the formation of autophagosomes. The inhibition of autophagy using Baf A1 led to a greater accumulation of p62 and LC3I in cells treated with LQ. However,

**Fig. 1 Glutaminolysis sustains the production of ATP to inhibit AMPK and to activate mTORC1. A** ATP/ADP ratio of U2OS cells incubated in the presence or the absence of all amino acids for 2 or 72 h. Fed cells (C) are used as control. **B** ATP/ADP ratio of U2OS cells incubated in absence of amino acid for the indicated time. **C** ATP/ADP ratio of amino acid-starved U2OS cells incubated in the presence or absence of LQ during the indicated times. **D** OCR analysis by Seahorse® technology of amino acid-starved U2OS cells incubated in the presence (blue) or absence (purple) of LQ during 72 h. OCR was measured either in basal conditions or after the injection of oligomycin, FCCP, and rotenone/antimycin A. Data are mean ± SEM of three biologically independent experiments performed with five replicates. **E** Basal respiration used to drive ATP production as determined by OCR quantification of data obtained in (**D**). **F** Immunoblot of mTORC1 activity markers (S6K, S6, and 4EBP1 phosphorylation) and AMPK phosphorylation of U2OS cells incubated with or without amino acids, in the presence or absence of LQ during 72 h. **G** Immunoblot of mTORC1 activity markers (S6K and S6 phosphorylation) and AMPK phosphorylation of amino acid-starved U2OS cells incubated in the presence or absence of LQ during 24, 48, or 72 h. **H** Immunofluorescence microscopy captions of U2OS cells incubated with or without amino acids, in the presence or absence of LQ during 72 h. Cells were stained against LAMP2 (lysosomal marker, red), mTORC1 (green) and DAPI (blue). Scale bar represents 10 μm. **I** Quantification of the colocalization between LAMP2 and mTORC1 as shown in (**H**). Person's R value was evaluated using ImageJ coloc2 plugin on 25 ROI in three biologically independent experiments (75 ROI in total per condition). **J** Immunoblot of mTORC1 activity markers (S6K and S6 phosphorylation) and AMPK phosphorylation of amino acid-starved U2OS cells incubated in the presence or absence of LQ, with or without AICAR, during 72 h. **K** Immunoblot analysis of mTORC1 activity marker (S6 phosphorylation) of U2OS cells expressing a myc-tagged, constitutively active AMPK mutant in the presence or absence of amino acids and LQ as indicated. **L** Immunoblot of autophagy (p62 and LC3-I/II) and mTORC1 (S6K and S6 phosphorylation) markers of amino acid-starved U2OS cells incubated in absence or presence of LQ, AICAR, metformin or A769662 for 72 h, as indicated. **M** Fluorescence microscopy captions of GFP-LC3 expressing amino acid-starved U2OS cells incubated in the presence or absence of LQ, with or without AICAR, metformin or A769662 for 72 h. Autophagosome formation upon GFP-LC3 aggregation was assayed using confocal microscopy. The scale bar represents 10 μm. **N** Quantification of the number of GFP-LC3 dots per cell of captions obtained in (**M**). >100 cells were counted per experiment. **O** Immunoblot analysis of autophagy markers (p62 and LC3I/II) of U2OS cells treated with LQ, AICAR and/or Bafilomycin A1 as indicated for 72 h. Graphs show mean values ± SEM ($n = 3$ biologically independent experiments). *$p < 0.05$ (ANOVA analysis followed by a post hoc Bonferroni test). Source data are provided as a Source Data file.

reactivation of AMPK using AICAR concomitantly to Baf A1 and LQ did not accumulate p62 at the same level (Fig. 1O), and restored the levels of LC3II. Altogether, these results showed that long-term LQ treatment was sufficient to increase the levels of ATP, leading to the inhibition of AMPK. AMPK inhibition in LQ-treated cells led to the subsequent activation of mTORC1 and the inhibition of autophagy, two major processes controlling glutamoptosis.

**AMPK inhibition is necessary for glutamoptosis both in vitro and in vivo.** Next, we assessed the role of AMPK in the induction of apoptosis following the activation of mTORC1 by glutaminolysis during nutritional imbalance, i.e., glutamoptosis. In agreement with our previous results[12], long-term LQ treatment in absence of any other amino acid decreased cell viability (Fig. 2A, B). The LQ-induced increase in cell death was prevented by AMPK reactivation using AICAR, metformin, or A769662 (Fig. 2A, B). Further, a clonogenic assay confirmed that the pharmacological reactivation of AMPK prevented LQ-mediated cell death induction (Fig. 2C, D). AMPK reactivation was also sufficient to prevent apoptosis induced by LQ, as determined by the decrease in the double annexin V/PI staining observed by flow cytometry (Fig. 2E, F), and by the decrease in the pro-apoptotic markers cleaved PARP and cleaved caspase 3 (Fig. 2G). On the contrary, AMPK ablation did not increase the induction of cell death during amino acid withdrawal, as mTORC1 inhibition and subsequent autophagy activation under these conditions prevented glutamoptosis (Supplementary 2A). This result further positioned mTORC1/autophagy downstream of AMPK during glutamoptosis induction. These results were confirmed using ATG5$^{-/-}$ MEFs in which the impairment of autophagy completely abolished the capacity of AICAR to promote cell survival during amino acids starvation (Fig. 2H, I) as autophagy is necessary as survival mechanism. Thus, AMPK inhibition upstream of mTORC1/autophagy was necessary for glutamoptosis in vitro.

The participation of AMPK in glutamoptosis was also investigated in vivo using implanted tumors in xenograft mouse models. Two different cohorts of mice were implanted in the right dorsal flank with HCT116 cells. Mice were treated with either vehicle or with a cell-permeable αKG derivative, dimethyl-αKG

(DMKG) to induce glutamoptosis in vivo. Mice were then co-treated with the mTORC1 inhibitor temsirolimus (a rapamycin derivative) or with metformin. First, the induction of mTORC1 by DMKG was assessed by the analysis of S6 phosphorylation in these tumors. As expected, the injection of DMKG in mice induced the activation of mTORC1. A decrease of S6 phosphorylation and thus mTORC1 activation was observed in mice co-treated with temsirolimus or metformin (Fig. 2J, K and Supplementary 2B). Second, the in vivo induction of glutamoptosis was assessed by immunohistochemical analysis of cleaved caspase 3. We confirmed that DMKG treatment induced an increase in apoptotic cell death in xenograft tumors in vivo, as assessed by caspase 3 cleavage (Fig. 2L–M and Supplementary 2C). This is a validation of the physiological relevance of glutamoptosis in vivo, confirming that glutamoptosis can be used as a potential tool to induce tumoral cell death in vivo. Furthermore, as shown in Fig. 2L–M and Supplementary 2C, we also confirmed that the inhibition of mTORC1 using temsirolimus prevented glutamoptosis in vivo, similar to what we observed in cultured cells[12]. This result validated the physiological model by which the aberrant activation of mTORC1 during nutritional imbalance induces apoptotic cell death in animal models. Finally, and of note, AMPK reactivation using metformin which was confirmed by IHC on T172 AMPK phosphorylation (Supplementary 2D, E) was similarly sufficient to prevent glutamoptosis in vivo (Fig. 2L–M and Supplementary 2C). Altogether, these data confirmed the necessity of AMPK inhibition for glutamoptosis induction both in vitro and in vivo.

**Glutaminolysis is not necessary for LQ-mediated ATP production.** After confirming the capacity of LQ to sustain ATP levels, we investigated the possible role of glutaminolysis, the main pathway to catabolise glutamine, in the production of ATP in response to LQ. For this purpose, and following a loss-of-function approach, we inhibited GLS either pharmacologically or genetically (using two different inhibitors, BPTES and DON, or RNA-interference mediated knockdown). We then investigated the capacity of LQ treatment to induce ATP levels in GLS-inhibited cells. Surprisingly, the inhibition of GLS did not prevent the production of ATP in response to LQ in U2OS, HCT116 or HEK293 cells (Fig. 3A, B and Supplementary 3A, B). Confirming

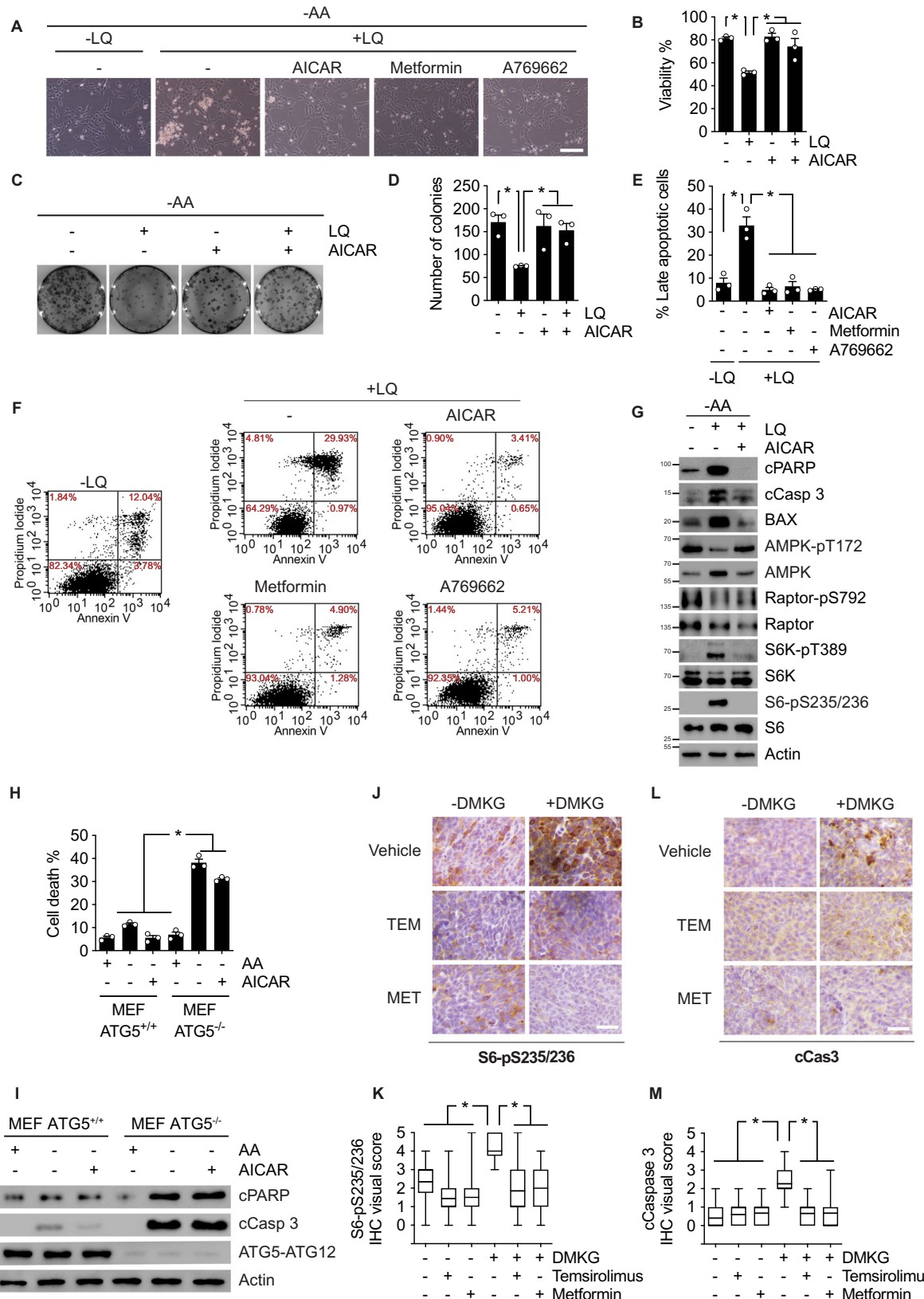

this result, silencing GDH did not impair the production of ATP in LQ-treated cells either (Fig. 3C). Thus, glutaminolysis did not show a necessary role for the capacity of LQ to sustain ATP levels. Further confirming this conclusion, we also observed that the addition of DMKG to amino acid-starved cells did not sustain the production of ATP to a similar extent than LQ treatment (Fig. 3D

and Supplementary 3C, D). Hence, glutaminolysis was neither necessary nor sufficient to sustain ATP production. The lack of decrease on ATP levels upon glutaminolysis inhibition correlated with an absence of AMPK reactivation (Fig. 3E, F), confirming that glutaminolysis did not mediate the connection between LQ and the ATP/AMPK axis.

**Fig. 2 AMPK inhibition is necessary for glutamoptosis both in vitro and in vivo. A** Representative microscopy images of amino acid-starved U2OS cells incubated in absence or presence of LQ, AICAR, metformin, or A769662 for 72 h, as indicated. Scale bar represents 100 μm. **B** Cell viability as estimated by a trypan blue exclusion assay of amino acid-starved U2OS cells incubated in the presence or absence of LQ and AICAR during 72 h. **C** Representative images of clonogenic assay of U2OS cells treated as in (**B**). **D** Colony quantification of images obtained in (**C**). **E** Quantification of late apoptosis population of three biologically independent experiments (double positive, annexin V and PI) as obtained in (**F**) for the indicated condition. **F** Flow cytometry analysis of annexin V/PI staining of amino acid-starved U2OS cells incubated with or without LQ in combination with AICAR, metformin, or A769662 during 72 h. **G** Immunoblot of the pro-apoptotic markers (BAX, cleaved caspase 3, and cleaved PARP), mTORC1 activity markers (Raptor, S6K, and S6 phosphorylation), and AMPK phosphorylation of amino acid-starved U2OS cells incubated in the presence or absence of LQ, with or without AICAR, during 72 h. **H** Cell viability as estimated by a trypan blue exclusion assay of ATG5$^{+/+}$ and ATG5$^{-/-}$ MEFs incubated in the presence or absence of amino acids (AA) and AICAR during 72 h. **I** Immunoblot analysis of pro-apoptotic markers of ATG5$^{+/+}$ and ATG5$^{-/-}$ MEFs incubated as in (**H**). **J–L** Representative immunohistochemistry microscopy pictures (×40 magnification) of xenograft tumors of mice treated as indicated (TEM: Temsirolimus; MET: Metformin). Samples were stained against S6-pS235/236 (**J**) and cleaved caspase 3 (**L**). Scale bars represent 100 μm. **K–M** IHC visual score of S6-pS235/236 (**K**) and caspase 3 (**M**) of images from (**J**) and (**L**), respectively. The upper and lower limits of the boxes represent quartiles, with the line within the boxes indicating the median and the whiskers showing the extremes ($n \geq 10$ images per treatment). Graphs show mean values ± SEM ($n = 3$ biologically independent experiments). *$p < 0.05$ (ANOVA analysis followed by a post hoc Bonferroni test). Source data are provided as a Source Data file.

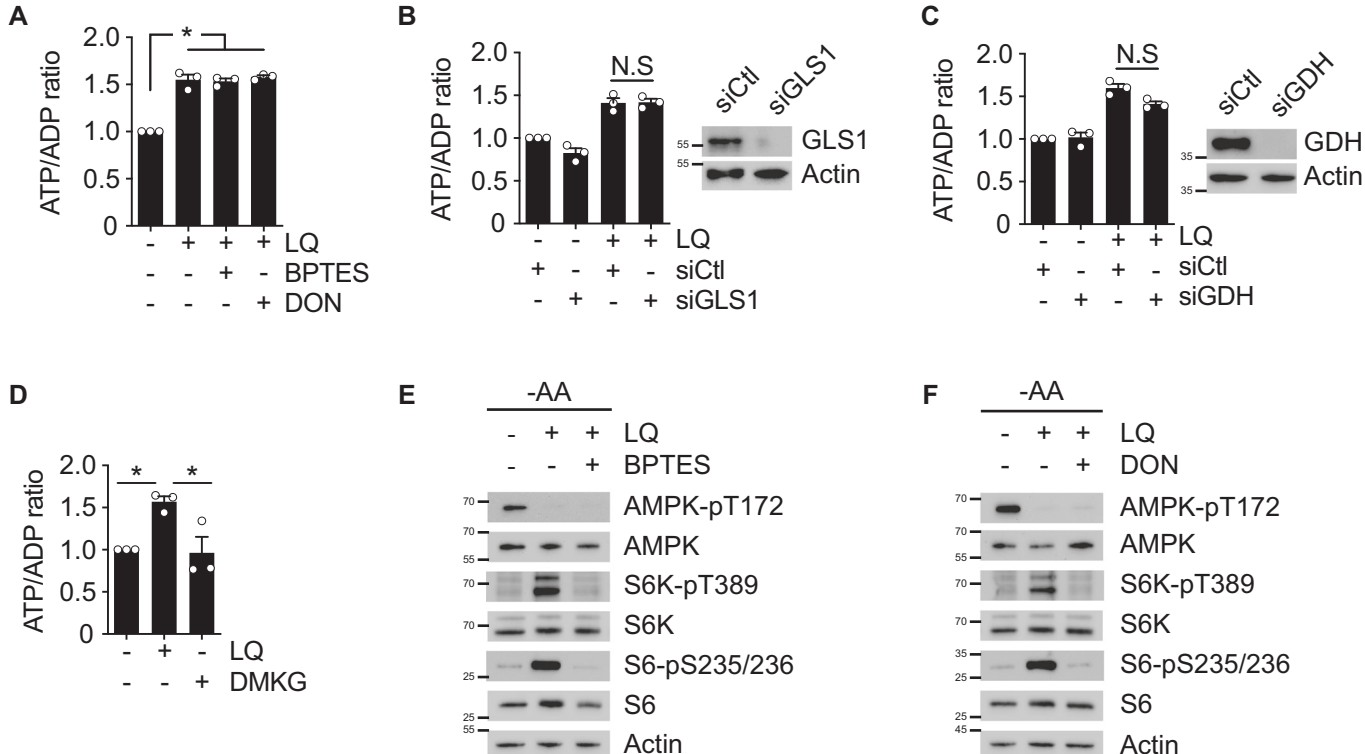

**Fig. 3 Glutamine is sufficient to sustain ATP levels, but not to activate mTORC1. A** ATP/ADP ratio of amino acid-starved U2OS cells incubated in the presence or absence of LQ with or without BPTES or DON during 72 h. **B, C** GLS or GDH expressions were knocked down using small interfering RNA (siRNA) in U2OS cells for 48 h. Cells were then treated with LQ for 72 h and the ATP/ADP ratio was measured. Scramble non-targeting siRNA was used as a control. Immunoblots of GLS or GDH levels are presented as a control of the knockdown. **D** ATP/ADP ratio of amino acid-starved U2OS cells treated with LQ or DMKG for 72 h. **E, F** Immunoblot of amino acid-starved U2OS cells treated with or without LQ in combination with BPTES (**E**) or DON (**F**) for 72 h. Activity markers of AMPK and mTORC1 were analysed. Graphs show mean values ± SEM ($n = 3$ biologically independent experiments). *$p < 0.05$ (ANOVA analysis followed by a post hoc Bonferroni test). Source data are provided as a Source Data file.

**Glutamine is sufficient to sustain ATP levels, but not to activate mTORC1.** During LQ treatment, leucine addition plays an allosteric activation role for GDH, necessary for αKG production[5,21]. As we concluded that glutaminolysis did not affect the ATP/AMPK pathway, we then assessed whether leucine had any role in ATP levels during LQ treatment. For this purpose, we treated amino acid-starved U2OS, HCT116 and HEK293A cells with glutamine or leucine, and subsequently measured ATP levels. Unlike leucine, glutamine alone was sufficient to sustain ATP levels in all three cell lines (Fig. 4A and Supplementary 4A, B). Confirming this result, we observed that basal respiration of the cells in amino acid starvation did not change in response to leucine addition (Fig. 4B). However, cells in the presence of glutamine showed a significant increase of the respiratory activity, independently of the presence or the absence of leucine (Fig. 4B). Similarly, glutamine sufficiency induced a clear increase of the OCR linked to ATP production by the mitochondria to meet the energetic needs of the cell (Fig. 4C). This increase of ATP levels by glutamine alone also correlated with an increase of endoplasmic reticulum calcium pool due to the activity of the sarcoendoplasmic reticulum Ca$^{2+}$ ATPase (SERCA) (Supplementary 4C). In a similar manner, and correlating with ATP levels, a strong decrease of AMPK phosphorylation was observed upon glutamine addition to otherwise amino acid-depleted cells, while the addition of leucine did not reduce the phosphorylation status of AMPK (Fig. 4D and Supplementary 4D).

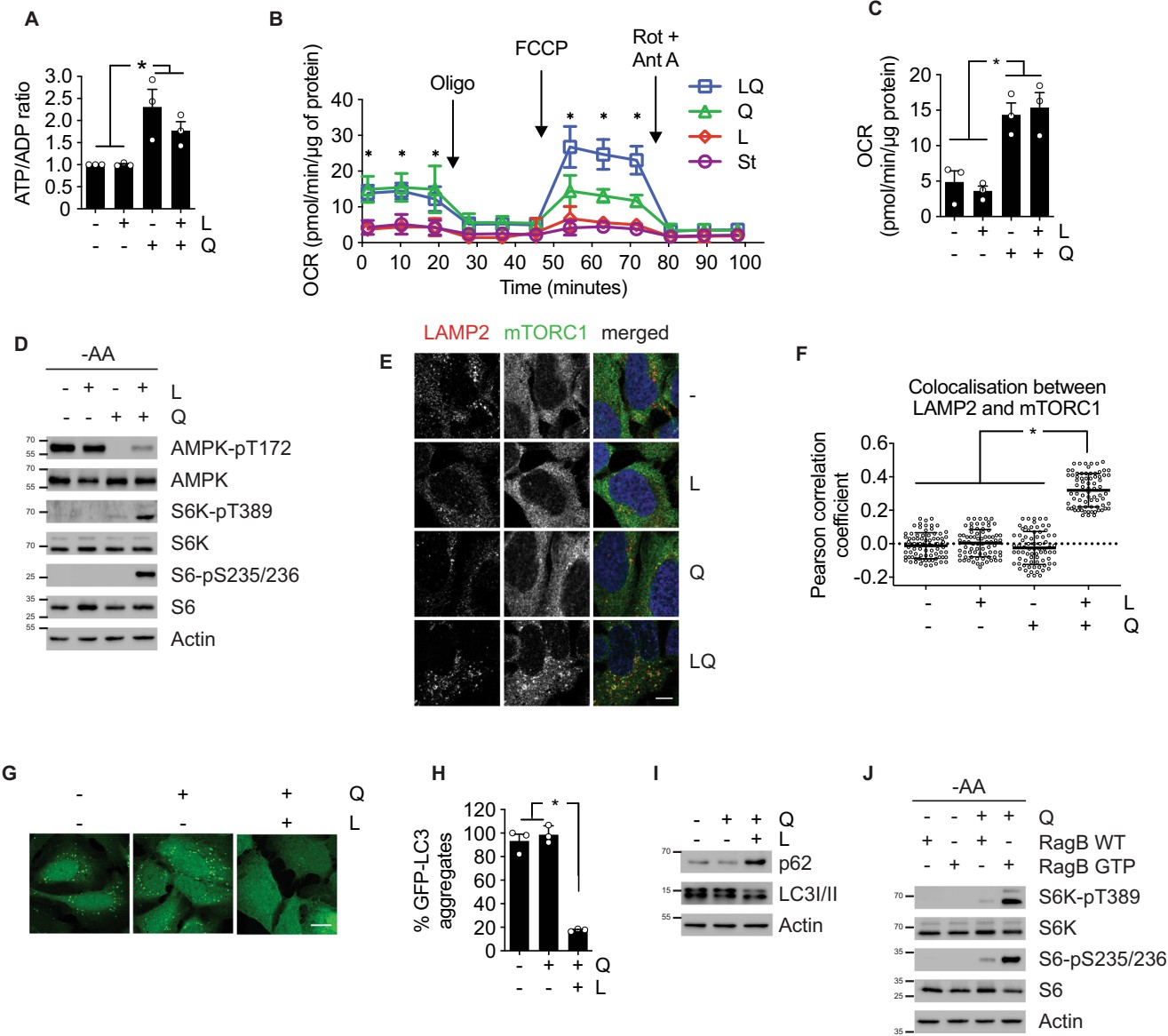

**Fig. 4 Glutamine metabolism activates mTORC1 following two parallel, necessary branches. A** ATP/ADP ratio of amino acid-starved U2OS cells incubated with leucine and/or glutamine during 72 h. **B** OCR analysis by Seahorse® technology of amino acid-starved (purple) U2OS cells incubated with leucine (red) and/or glutamine (Q green, LQ blue) during 72 h. OCR was measured either in basal conditions or after the injection of oligomycin, FCCP, and rotenone/antimycin A. Data are mean ± SEM of three biologically independent experiments performed with five replicates. **C** Basal respiration used to drive ATP production as determined by OCR quantification of data obtained in (**B**). **D** Immunoblot of mTORC1 activity markers (S6K and S6 phosphorylation) and AMPK phosphorylation of amino acid-starved U2OS cells incubated with leucine and/or glutamine during 72 h. **E** Immunofluorescence microscopy captions of U2OS cells incubated with leucine and/or glutamine during 72 h. Cells were stained against LAMP2 (lysosomal marker, red), mTORC1 (green) and DAPI (blue). Scale bar represents 10 μm. **F** Quantification of the colocalization between LAMP2 and mTORC1 as shown in (**E**). Person's R value was evaluated using ImageJ coloc2 plugin on 25 ROI in three biologically independent experiments (75 ROI in total per condition). **G** Fluorescence microscopy captions of GFP-LC3 expressing amino acid-starved U2OS cells incubated in the presence of glutamine and/or leucine during 72 h. Autophagosome formation upon GFP-LC3 aggregation was assayed using confocal microscopy. The scale bar represents 10 μm. **H** Quantification of the number of GFP-LC3 dots per cell of captions obtained in (**G**). >100 cells were counted per experiment. **I** Immunoblot of autophagy (p62 and LC3-I/II) markers of amino acid-starved U2OS cells incubated in the presence of glutamine and/or leucine during 72 h. **J** U2OS cells were transfected with RagB WT plasmid or RagB 54 L plasmid. Amino acid-starved cells were then incubated with or without glutamine for 72 h. Downstream targets of mTORC1 (S6K and S6 phosphorylation) were assessed by immunoblot. Graphs show mean values ± SEM ($n = 3$ biologically independent experiments). *$p < 0.05$ (ANOVA analysis followed by a post hoc Bonferroni test). Source data are provided as a Source Data file.

Similar to what was observed for LQ treatment (Fig. 3B, C), gluta-minolysis inhibition by GLS1 or GDH knockdown did not affect the capacity of glutamine sufficiency to induce ATP levels in U2OS and HCT116, confirming that glutaminolysis was not necessary for glutamine-dependent ATP production (Supplementary 4E–H). It is worth noting that neither glutamine alone nor leucine alone (in

absence of any other amino acid) were sufficient to activate mTORC1 after 72 h in either U2OS or HCT116 cells (Fig. 4D and Supplementary 4D). Consistently, mTORC1 localization at the surface of the lysosome was only observed in the presence of both leucine and glutamine, but it was not observed if only one of these amino acids were added (Fig. 4E, F and Supplementary 4I, J). In a similar manner

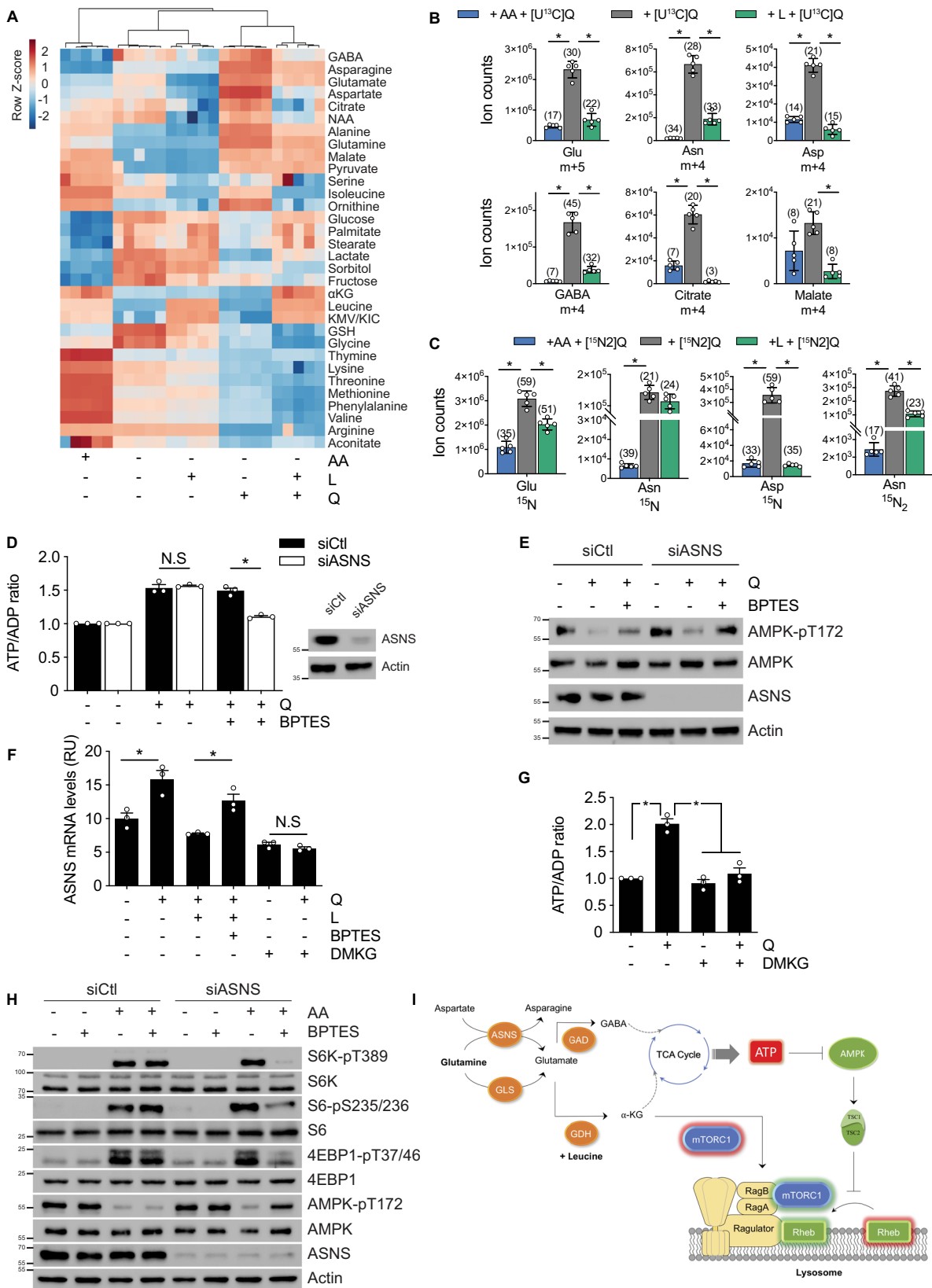

than for LQ, glutamine levels were measured in the cell culture medium during glutamine sufficiency condition to verify the availability of this amino acid even after 3 days of treatment (Supplementary 4K). Autophagy analysis downstream of mTORC1 also confirmed the inability of glutamine sufficiency to activate mTORC1. Thus, long-term glutamine addition did not inhibit

autophagy activation during amino acid starvation, as assessed by the autophagy reporter GFP-LC3 by confocal microscopy, and by immunoblot analysis of the autophagic endogenous markers p62 and LC3I/II. In contrast, and as shown previously, the addition of both leucine and glutamine strongly decreased the number of GFP-LC3 aggregates and increased p62 levels in U2OS cells (Fig. 4G–I).

**Fig. 5 ASNS and GABA shunt are alternative pathways to metabolize glutamine. A** Heatmap representation of metabolite levels, as determined by LC–MS analysis, in amino acid-starved U2OS cells incubated with glutamine and/or leucine. The heatmap was created with MetaboAnalyst3.0 with the total pools of the detected metabolites. **B** [13]C-labeled metabolite levels, as determined by LC–MS analysis, in U2OS cells incubated with or without all amino acids or leucine alone as indicated, in the presence of (U)-[13]C-glutamine during 72 h. Total ion counts of glutamate m + 5, asparagine m + 4, aspartate m + 4, GABA m + 4, citrate m + 4, and malate m + 4 are graphed. The percentage of labeling with respect to total metabolite levels is shown in parenthesis for each metabolite. **C** [15]N-labeled metabolite levels, as determined by LC–MS analysis, in U2OS cells incubated with or without all amino acids or leucine alone as indicated, in the presence of [15]N2-glutamine during 72 h. Ion counts of [15]N -glutamate, [15]N -asparagine, [15]N aspartate, and [15]N2-asparagine are plotted. The percentage of labeling relative to the total metabolite is shown in parenthesis for each metabolite. **D** ASNS expression was knocked down using small interfering RNA (siRNA) in U2OS cells during 48 h. Cells were then treated with glutamine and BPTES as indicated for 72 h and the ATP/ADP ratio was measured. Scramble non-targeting siRNA was used as a control. Immunoblot of ASNS levels is presented as a control of the knockdown. **E** Immunoblot analysis of AMPK phosphorylation in U2OS cells treated as in (**D**). **F** Relative mRNA expression levels of ASNS as determined by qPCR in amino acid-starved U2OS cells incubated as indicated during 72 h. **G** ATP/ADP ratio of amino acid-starved U2OS cells incubated with glutamine and/or DMKG during 72 h. **H** Immunoblot analysis of mTORC1 downstream target (S6K, S6, and 4EBP1 phosphorylation) and AMPK phosphorylation in U2OS cells incubated during 24 h in the presence of all amino acids with dual inhibition of GLS and/or ASNS. ASNS expression was knocked down using small interfering RNA (siRNA) in U2OS cells during 48 h. Scramble non-targeting siRNA was used as a control. Cells were then incubated in the presence of absence of all amino acids and/or BPTES as indicated during 24 h. **I** Schematic representation of the two branches model connecting glutamine metabolism and mTORC1 signaling. Graphs show mean values ± SEM ($n = 3$ biologically independent experiments). *$p < 0.05$ (ANOVA analysis followed by a post hoc Bonferroni test). Source data are provided as a Source Data file.

It is well documented that the re-addition of leucine alone for 10–30 m after a short-term amino acid starvation (1–2 h) is sufficient to activate mTORC1[22–25]. However, our data showed that this reactivation of mTORC1 disappeared after longer starvation periods (>4 h of amino acid starvation) (Supplementary 4L), which correlated with a strong decrease of glutamine intracellular levels following 4 h of amino acid starvation (Supplementary 4M). In contrast, re-addition of both leucine and glutamine activated mTORC1 even after longer periods of amino acid starvation (Supplementary 4N). These results support a model by which leucine alone activates mTORC1 after short-term amino acid starvation thanks to the remaining pool of intracellular glutamine. It has been proposed elsewhere that intracellular glutamine is required for leucine uptake[26]. However, our data showed that, in our setup, intracellular levels of leucine were not significantly affected by the co-addition of glutamine in starved cells (Supplementary 4O), suggesting that intracellular glutamine was not necessary for leucine uptake. Rather, our data sustained a model by which leucine only activates mTORC1 by cooperating with glutamine to activate glutaminolysis.

**Glutamine metabolism activates mTORC1 following two parallel, necessary branches.** In light of our results, we investigated the possibility that two parallel pathways connect glutamine and mTORC1: a previously reported glutaminolysis-dependent branch; and an ATP-dependent but GLS/GDH-independent branch. We previously showed that glutaminolysis activates mTORC1 via RagB. Thus, αKG generation by glutaminolysis increases GTP loading of RagB, leading to mTORC1 translocation to the surface of the lysosome[5]. It is documented that the overexpression of a constitutively GTP-bound mutant variant of RagB (RagB-GTP mutant) is sufficient to force the translocation of mTORC1 to the surface of the lysosome even in amino acid-starved cells[27]. The expression of this RagB-GTP mutant is sufficient to activate mTORC1 in amino acid-starved cells at short times (1–2 h)[5,27]. However, we observed that RagB-GTP mutant expression did not activate mTORC1 in amino acid-starved cells at longer times (72 h, Fig. 4J and Supplementary 4P). Strikingly, the combination of both RagB-GTP expression and glutamine addition induced the full activation of mTORC1 at 72 h, as determined by the phosphorylation of the downstream targets S6K and S6 (Fig. 4J). Similar results were obtained using a RagD-GDP mutant, which also upregulates the lysosomal translocation of mTORC1 (Supplementary 4P). Opposite to what we observed with glutamine, the combination of both RagB-GTP and leucine

did not induce the activation of mTORC1 signaling (Supplementary 4Q). These results further sustained a model by which glutaminolysis (activated by both glutamine and leucine) is necessary to produce αKG and to induce the Rag-mediated translocation of mTORC1 at the surface of the lysosome. On the other hand, glutamine, independently of GLS and GDH, sustains the production of ATP and the inhibition of AMPK, necessary for the full activation of mTORC1 at the surface of the lysosome, mediated by the mTORC1 coactivator Rheb[28].

**ASNS and GABA shunt are alternative pathways to metabolize glutamine.** So far, our results indicated that glutamine sufficiency produces ATP through mitochondrial activity, but glutaminolysis was not necessary for this. To uncover the pathways involved in glutamine-induced ATP production, we performed a metabolomics analysis (Fig. 5A). We observed that, during amino acid starvation (72 h), the levels of a large number of metabolites dropped. Addition of leucine did not restore the vast majority of them, again confirming that leucine has mostly an allosteric role, not having a significant impact on the metabolism of the cell by itself. In contrast, the addition of glutamine changed completely the pattern, allowing us to identify a group of metabolites which levels raised very high, particularly GABA, asparagine, glutamate and aspartate (Fig. 5A and Supplementary 5A–F). This result indicated an increased synthesis of asparagine, mediated by the activity of ASNS, responsible for the production of asparagine and glutamate from glutamine and aspartate (Supplementary 5G). To experimentally test this possibility, we traced glutamine during glutamine sufficiency in U2OS cells using [U[13]C]-glutamine. As expected, we observed an accumulation of both glutamate m + 5 and asparagine m + 4 in the intracellular extract of U2OS cells after 72 h of treatment. Further, glutamine sufficiency also produced m + 4 labeled aspartate (Fig. 5B), indicating that aspartate used by ASNS was indeed originated (recycled) from the glutamate through oxaloacetate generation at the TCA cycle. Glutamate produced by ASNS was further converted into aspartate likely through the activity of the enzyme aspartate transaminase (GOT1) (Supplementary 5G). In parallel, we also incubated amino acid-starved cells in the presence of [[15]N2]-glutamine for 72 h. In agreement with our previous observations, we observed a large accumulation of [15]N-asparagine, confirming the implication of ASNS in the metabolization of glutamine during glutamine sufficiency (Fig. 5C). The production of both [15]N-aspartate and [15]N-asparagine in this condition finally

confirmed the involvement of GOT1 for the re-cycling of aspartate from glutamate. Of note, the addition of leucine to glutamine reduced the production of $^{15}$N-aspartate, confirming that in the presence of leucine, glutamate is mostly catabolized through GDH, and not through GOT1.

GABA levels were also significantly increased during glutamine sufficiency (Supplementary 5F). Following a similar approach than before, we observed that U2OS cells cultured with [U$^{13}$C]-glutamine showed high levels of m+4 labeled GABA (Fig. 5B), confirming that GABA was produced directly from glutamine. As for aspartate, a significant decrease of GABA levels was observed in LQ treatment with respect to glutamine alone, again confirming the potent allosteric role of leucine diverting glutamine catabolism through GDH.

The results shown above suggest that ASNS participates in a metabolic pathway, alternative to glutaminolysis, to metabolize glutamine in the absence of leucine. Therefore, to link ASNS with the production of ATP during glutamine sufficiency, we performed a loss-of-function approach with ASNS and we measured the levels of ATP in glutamine-treated cells. Indeed, while neither the genetic inhibition of ASNS (gene silencing) nor the pharmacological inhibition of GLS (using BPTES) affected ATP levels, the inhibition of both GLS and ASNS at the same time prevented the capacity of glutamine sufficiency to induce ATP levels in both U2OS and HCT116 cells (Fig. 5D and Supplementary 5H). Downstream of ATP, the co-inhibition of both ASNS and GLS reactivated AMPK phosphorylation even in the presence of glutamine, while it did not impact mTORC1 signaling (Fig. 5E and Supplementary 5I). These results confirmed that both GLS and ASNS were sufficient to metabolize glutamine, sustaining the ATP/AMPK pathway during glutamine sufficiency, one compensating for each other. Explaining the involvement of ASNS in glutamine metabolism in the absence of the allosteric input of leucine (i.e., when glutaminolysis was inactive), we observed that the expression levels of ASNS were high during glutamine sufficiency, but ASNS mRNA levels decreased when leucine was co-added together with glutamine (Fig. 5F). Similarly, ASNS mRNA levels decreased significantly in cells incubated in the presence of DMKG, suggesting that αKG (the end-up product of glutaminolysis) was ultimately responsible for the inhibition of ASNS expression. The reduction of ASNS mRNA levels by LQ was blocked by preventing αKG production using BPTES (Fig. 5F), further confirming the role of αKG in the control of ASNS expression. This observation correlated with the inhibition of ATP production by glutamine when the cells are concomitantly treated with DMKG (Fig. 5G). These results suggested that glutaminolytic αKG prevents the metabolization of glutamine by ASNS. Conversely, when glutaminolysis is not operating, the absence of αKG may allow the alternative conversion of glutamine into asparagine and glutamate by ASNS to guarantee ATP production in the cell. The alternative functioning of ASNS and GABA shunt, operating only when glutaminolysis is inactive, was further confirmed by the observation that silencing ASNS or GAD (the enzyme responsible for the first reaction of the GABA shunt) only affected LQ-induced mTORC1 and cell death in the presence of BPTES (Supplementary 5J, K), confirming that both ASNS and GABA shunt are dispensable when glutaminolysis is active.

**GLS and ASNS cooperate to connect glutamine and mTORC1 signaling**. The connection between glutamine metabolism and mTORC1 is a major feature of cancer cells. Glutaminolysis inhibition has been advocated as a potential approach to target tumor growth through mTORC1 inhibition. However, GLS inhibitors have failed to show a significant effect as

therapeutic drugs[29,30]. Indeed, in vitro GLS inhibition is not able to inhibit mTORC1 in the presence of all amino acids, suggesting that alternative pathways are linking glutamine metabolism and mTORC1 activation. The results presented above identified asparagine synthesis as a key pathway for the catabolism of glutamine. Hence, we investigated if GLS and ASNS are two necessary branches for the activation of mTORC1 in response to nutrients. For this purpose, we investigated the impact of the dual inhibition of ASNS and GLS on mTORC1 activation in the presence of all amino acids for 24 h in U2OS cells, using respectively siRNA knocking down and BPTES. As observed before, the inhibition of GLS using BPTES was unable to inhibit mTORC1 pathway (Fig. 5H). Likewise, ASNS downregulation did not have a significantly impact in the phosphorylation of mTORC1 downstream targets (S6K, S6, and 4EBP1), confirming that GLS and ASNS can compensate for each other's activities. However, and confirming the model of two independent branches, the dual inhibition of ASNS and GLS significantly decreased mTORC1 activity in the presence of all amino acids (Fig. 5H). Consistently, while neither BPTES treatment nor ASNS silencing could restore AMPK phosphorylation in the presence of all amino acids, we observed an almost complete restoration of AMPK phosphorylation upon the dual inhibition of GLS (BPTES treatment) and ASNS knockdown (Fig. 5H). This result was in line with the inhibition of mTORC1 observed in these conditions. The need for a dual GLS and ASNS inhibition could explain the lack of effectivity of targeting only glutaminolysis as a therapeutic approach against cancer. This conclusion led us to propose ASNS and GLS as dual targets that should be inhibited to allow an efficient inhibition of mTORC1 in cancer cells.

## Discussion

The results presented herein describe a dissection of the molecular mechanisms connecting glutamine metabolism and mTORC1 signaling. Previously, we had demonstrated the sufficiency of glutamine metabolism, particularly glutaminolysis, to translocate mTORC1 to the surface of the lysosome, where it is subsequently activated[27]. However, until now the role of the bioenergetic status of the cell and AMPK signaling in the glutamine-to-mTORC1 connection was never clarified. Here, we show that the full activation of mTORC1 downstream of glutamine strictly requires ATP production and AMPK inhibition. In this report we also demonstrated the implication of the ATP/AMPK axis in the induction of glutamoptosis, i.e., the unusual apoptotic cell death induced upon the inhibition of autophagy during nutritional imbalance[12]. Our results, furthermore, confirmed the glutamoptosis concept in mouse models, validating nutritional imbalance as a potential strategic approach to induce tumor cell death in vivo.

Contrary to our first hypothesis, the connection between glutamine and ATP/AMPK axis did not require glutaminolysis, but instead followed an alternative pathway involving the catalytic activity of ASNS, and GABA shunt. All these three elements have been connected previously to cancer metabolism and therapy resistance. However, this is the first time that this alternative pathway has been shown to mediate the connection between glutamine and mTORC1 signaling. Although in a different context, our results confirmed previous observations identifying ASNS as a mediator of mTORC1 activation by the amino acid asparagine[7,31]. Now, our results supported that ASNS metabolizes glutamine, and in combination with GOT1 and GABA shunt, provides with an anaplerotic entry at the TCA cycle for the production of oxaloacetate and the subsequent ATP synthesis in the absence of glutaminolysis. It is important to emphasize that this alternative pathway operates only when glutaminolysis is not

active, as glutaminolysis seems to be the most efficient mechanism to feed the TCA cycle from glutamine.

Furthermore, the existence of an alternative pathway connecting glutamine and mTORC1 might explain the lack of efficacy of targeting GLS as a potential strategy against cancer. Our results demonstrated that, even in the absence of GLS activity, cancer cells maintain the capacity to metabolize glutamine through ASNS, which expression in these conditions is increased due to the reduction in αKG production.

Although the role of leucine in mTORC1 activation has been extensively described, most of the leucine-induced activation mechanism for mTORC1 signaling have been described following short-term approaches (10 min–2 h)[22,25]. The accepted model for mTORC1 activation in response to amino acids[32] assigns to leucine a major role in mTORC1 activation, with additional amino acids (such as glutamine and arginine) playing partial activation mechanisms[25,31]. Now, our results indicated that leucine does not show sufficiency for mTORC1 activation at long-term, showing a permissive role for glutamine to be metabolized through GDH for αKG production, as leucine is an allosteric activator of GDH. In our model, and contrary to what was proposed elsewhere[26], the intracellular levels of leucine did not change in response to glutamine availability. Still, our metabolomics, energetic, and signaling data indicated that leucine has almost no effect in cellular physiology in the absence of glutamine, severely questioning the currently accepted model placing leucine as the major signaling amino acid in terms of mTORC1 activation.

Our results support a model (Fig. 5I) in which glutamine is connected to mTORC1 through two parallel, necessary pathways: the GLS/GDH-mediated production of αKG that allows the translocation of mTORC1 to the lysosome downstream of Ragulator/RAG; and the ASNS-mediated production of ATP, that allows the inhibition of AMPK and the complete activation of mTORC1 at the surface of the lysosome. How this double pathway interacts with the lysosomal V-ATPase-mediated mechanism during mTORC1 activation would require further investigations.

## Methods

**Cell culture**. U2OS, HCT116, and HEK293A cell lines were obtained from the American Type Culture Collection (ATCC). GFP-LC3 expressing U2OS cells were kindly provided by Prof. Eyal Gottlieb (Cancer Research UK, Glasgow, UK). ATG$^{+/+}$ MEFs and ATG5$^{-/-}$ MEFs were kindly provided by Prof. Patricia Boya (Centro de Investigaciones Biologicas, Madrid, Spain). AMPK$^{+/+}$ MEFs and AMPK$^{-/-}$ MEFs were kindly provided by Prof. Benoit Viollet (Institute Cochin, Paris, France). All cell lines were cultured in a high glucose DMEM base medium (D6429 Sigma-Aldrich) supplemented with 10% v/v of inactivated foetal bovine serum, glutamine (2 mM), penicillin (100 IU ml$^{-1}$) and streptomycin (100 µg mL$^{-1}$) at 37 °C, 5% CO$_2$ in humidified atmosphere. All cell lines were regularly tested for mycoplasma contamination by PCR. Amino acid starvation was performed with EBSS medium (E2888 Sigma-Aldrich) complemented with glucose at a final concentration of 4.5 g L$^{-1}$, at the indicated times, ranging from 2 h to 72 h. When indicated, starvation medium was complemented at time 0 with glutamine (2 mM), leucine (0.8 mM) or DMKG (2 mM). Different inhibitors or activators were used concomitantly at the following concentrations: AICAR (0.5 mM), Metformin (5 mM), A769662 (100 µM), BPTES (29 µM), and DON (40 µM).

**Plasmids**. The following plasmids were obtained from Addgene: RagB wt (#19301), RagB 99 L (#19303), RagD wt (#19307), RagD 77 L (#19308) and HA (#128034). myc-CA-AMPK plasmid was kindly provided by Prof. Benoit Viollet (Institute Cochin, Paris, France). The plasmid transfections were carried out using Jetpei (Polyplus Transfection) according to the manufacturer's instructions. 70% confluent cells were transfected with the indicated amount of plasmid, 24 h later cells were treated as indicated.

**ATP/ADP ratio measurement**. Cells were seeded in a white 96 well flat bottom plate 1 day prior the treatment. After treatment the medium was removed, the cells washed twice with phosphate-buffered saline (PBS) and the measurement was performed using the luminometric ATP/ADP ratio assay kit (from Sigma-Aldrich MAK135) following the manufacturer's instruction. The luminescence was

measured using microplate reader TriStar² LB 942 (Berthold). The results are expressed as ratio relative levels of ATP on ADP.

**Immunoblotting**. U2OS or HCT116 cells were seeded and treated in 6 cm plates. After treatment, the cells were washed twice with PBS and protein were extracted using radioimmune precipitation assay (RIPA) lysis buffer (150 mM NaCl, 50 mM Tris, 1 mM EGTA, 1 mM EDTA, 1% (v/v) Triton X-100, 0.5% (w/v) sodium deoxycholate, 0.1% (v/v) SDS, pH 7.4) buffer containing a cocktail of protease inhibitor (P8340 Sigma), PMSF 1 mM, and inhibitors of phosphatases (P0044 and P5726 Sigma). Protein lysates concentrations were determined by Bradford assay (BCA kit from Thermo Fisher), normalized and 20 µg of protein were separated by SDS-PAGE electrophoresis. Transfer on nitrocellulose membranes was achieved using Trans-Blot Turbo (BioRad). Membranes were blocked in 5% BSA (TBS-Tween 20 0.01%) and incubated overnight with the indicated antibody. After incubation with the corresponding secondary antibody 2 h at room temperature, the membranes were finally imaged using the ChemiDoc MP image (BioRad).

**Flow cytometry**. Cells were seeded the day before treatment, treated for the indicated time and stained following with annexin V and propidium iodide (PI) (Annexin V early apoptosis detection kit, #6592 Cell Signaling Technology) according to the manufacturer's instructions. Finally, cells were analysed using BD FACSCalibur BD Biosciences flow cytometer. The analysis of the data was performed using the BD CellQuestTM Pro Analysis software.

**Real-time PCR**. The total messengers RNAs from cells were isolated using TRIzol reagent (Invitrogen). All samples displayed RIN greater than or equal to 8 as determined using the RNA 6000 Nano Kit and the bioanalyzer 2100 (Agilent Technologies). After quantification and normalization, 1 µg of total mRNA was retrotranscribed using GoScript Reverse Transcription System (Promega). Real-time PCR was performed in triplicate for each sample with SSO Advanced Universal SYBR Green Supermix (BioRad), 1 ng of cDNA and transcript-specific primers using Step-One Plus Real-Time PCR system (Applied Biosystems). Specific primers were purchased from Sigma-Aldrich. Specific primer sequences (purchased from Sigma-Aldrich) are given below and in the Supplementary Table.

ASNS forward primer (5′–3′) CAGCGGGGACCCAATAGTAG
ASNS reverse primer (5′–3′) GTGTAGGACGTGAGCAGAAAA

**Confocal microscopy**. Cells were cultured on coverslips for 24 h and treated for 72 h as indicated. Cells were then fixed with paraformaldehyde 4% for 30 m at room temperature. After fixation, cells are permeabilize 10 m using Triton-X100 0.05% and unspecific sites were blocked in BSA 5%—PBS for 30 m on a rocking platform at room temperature. Incubation with the primary antibodies was carried out at room temperature for 1 h and cells were incubated with the secondary antibody for another hour at room temperature. Finally, the coverslips were mounted on slides with Prolong containing DAPI (Invitrogen) and the samples were imaged with a LEICA confocal SP4 microscope. U2OS stably expressing LC3-GFP construct, were mounted directly after fixation in the same mounting medium and imaged with a widefield LEICA microscope.

**siRNA**. All siRNA were ON-TARGETplus SMARTpool (Dharmacon). The siRNA mediated knockdown was performed using 10 nM of siRNA transfected with INTERFERin® siRNA transfection reagent (POLYplus transfection) following the manufacturer's instructions. Cells were incubated in contact with the transfection agent for 48 h and then treated with the indicated conditions for 24, 48, 72, or 96 h.

**Clonogenic assay**. 70% confluent U2OS cells were starved in EBSS (4.5 g L$^{-1}$), treated with LQ and/or AICAR for 72 h. After treatment, $1.5 \times 10^3$ cells of each conditions were reseeded in 3 cm plates in complete medium. Fourteen days after, cells were fixed with paraformaldehyde 4% in PBS for 30 m and stained with crystal violet 5% solution.

**Oxygen consumption and extracellular acidification rate measurement**. To measure the OCR and extracellular acidification rate (ECAR) $5 \times 10^4$ cells were seeded the day before experiment in XFe24 Cell Culture microplate in high glucose DMEM base medium supplemented with 10% v/v of inactivated foetal bovine serum, glutamine (4 mM), penicillin (100 IU ml$^{-1}$) and streptomycin (100 µg mL$^{-1}$). The next day cells were washed twice in phosphate buffer saline (PBS) and medium was replaced with the HBSS buffered with hepes complemented or not with glutamine or/and leucine for 72 h. To eliminate residues of carbonic acid from medium, cells were incubated for at least 30 m at 37 °C with atmospheric CO2 in a non-humidified incubator. OCR and ECAR were assayed in a Seahorse XF-24 extracellular flux analyser by the addition via port A of 1 µM oligomycin (port A), port B of 2.5 µM carbonyl cyanide-p-trifluoromethoxyphenylhydrazone (FCCP), and port C of 1 µM rotenone and 1 µM antimycin A (port C). Three measurement cycles of 2-min mix, 2-min wait, and 4-min measure were carried out at basal condition and after each injection. At the end of the experiment, each well was washed twice with 50 µL of PBS and proteins were extracted with 50 µL of RIPA lysis buffer at room temperature.

Protein concentration in each well was measured by a BCA assay according to the manufacturer's instructions (Thermo).

**Liquid chromatography coupled to mass spectrometry (LC–MS) for metabolomics analysis**. For steady-state metabolomics or metabolite tracing, $7.5 \times 10^5$ U2OS cells were seeded in a six-well plate in the evening. The day after, the cells were washed twice with PBS and the medium was changed to EBSS (complemented with glucose to a final concentration of $4.5\,g\,L^{-1}$). All amino acid, glutamine, U-$^{13}$C-glutamine (Cambridge Isotope Laboratories, cat. no. CLM-1822-SP-PK), U-$^{15}$N$_2$-glutamine (Cambridge Isotope Laboratories, cat. no. NLM-1328-PK) or leucine were added as indicated. Cells were washed three times with PBS and the extraction buffer (50% methanol, 30% acetonitrile, 20% water, all LC–MS grade) was added ($1\,mL/1 \times 10^6$ cells). Cells were incubated in dry ice for 15 min, collected, vigorously shook for 15 min at 4 °C, and left 1 h at 20 °C. Samples were centrifuged at $15{,}000 \times g$ and supernatants were transferred to autosampler vials and stored at $-80$ °C until further analysis. To avoid bias due to machine drift, samples were randomized and processed blindly.

A Q Exactive mass spectrometer coupled to a Dionex U3000 UHPLC (both Thermo Fisher Scientific) system was used to perform the LC–MS analysis. A Sequant ZIC-pHILIC column ($150 \times 2.1$ mm, 5 μm) and guard column ($20 \times 2.1$ mm, 5 μm) (Merck Millipore) were utilized for the chromatographic separation[33]. The column oven temperature was maintained at 40 °C. The mobile phase was composed of 20 mM ammonium carbonate and 0.1% ammonium hydroxide in water (solvent A), and acetonitrile (solvent B). The flow rate was set at 0.2 ml/min with the following gradient: 80% B for 2 min, linear decrease to 20% of B 15 min. Both solvents were then brought back to initial conditions and maintained for 8 min. The mass spectrometer was operated in full MS and polarity switching mode.

The acquired spectra were analysed using XCalibur Qual Browser and XCalibur Quan Browser software (Thermo Fisher Scientific) by referencing to an internal library of compounds. Mass isotopologue distribution of metabolites was determined by integration of the corresponding peaks, and correction for natural abundance was performed using the Polly™ IsoCorrect tool from the cloud-based platform Elucidata (https://polly.elucidata.io).

**Xenograft mouse model**. All animals were bred in the Animal Facility A2 of the University of Bordeaux (institutional agreement number 133063916), led by Dr. Benoît Rousseau. The project received the agreement from the local Ethics Committee on Animal Experiments CEEA50 of Bordeaux (agreement number APA-FIS#10090-2017052409402562 v2). 8-week-old male NOD.Cg-Prkdc$^{scid}$ ll2rg$^{tm1Wjl}$/SzJ immunodeficient mice were randomly assigned to six different groups (10 mice per group). 2 million HCT116 cells were injected in 100 μL of medium into the right dorsal flank of the mouse subcutaneously. Tumor growth was monitored with a calliper three times a week. Intra-peritoneal treatment was initiated when the average volume of the tumors reached 200 mm³. Three times a week 200 μL of PBS, DMKG: 3 g/kg/week, Temsirolimus 10 g/kg/week and metformin 600 mg/kg/week were injected per mouse. After 3 weeks of treatment, mice were sacrificed, and the tumors were collected for immunohistochemistry (IHC) analysis.

**IHC**. Tissue sections, 3 mm thick, were prepared from formalin-fixed paraffin-embedded tissues and submitted to standard IHC protocols with rabbit anti-cleaved caspase-3 Asp175 (CS #9664), phospho-S6 Ribosomal Ser235/236 (CS#4856) or phospho-AMPKα (CS#2535). The peroxidase reaction product was visualized with 3.3′-diaminobenzidine tetrahydrochloride (DAB) reaction kit (Vector Lab, SK-4100). Omission of the primary antibody in the immunostaining procedure was used as a negative control. Sample analysis and images obtained with a Leica DM6000B microscope and Leica DFC500 digital camera.

**Cytosolic calcium**. $1 \times 10^5$ U2OS cells were seeded onto glass coverslip (25 mm) in DMEM. The day after, cells were treated for 48 h. Coverslips were mounted in a recording chamber positioned on the stage of an inverted epifluorescence microscope (IX70, Olympus) equipped with an ×40 UApo/340− 1.15 W objective. Single-cell cytosolic calcium imaging was performed, using Fluo2 LR-AM calcium (2 μM added in the medium 30 m before imaging). Fluo2-Leak Resistant (LR)-AM exhibits limited compartmentalization in intracellular stores and is leakage-resistant. Fluorescence intensity changes were normalized to the initial fluorescence value F0 and expressed as F/F0 (relative cytosolic Ca$^{2+}$ levels). One field was acquired from each coverslip and the data were pooled from three independent coverslips on three independent days. Data were processed using OriginPro 7.5 software (Origin Lab, Northampton, MA, USA). On graphs, data were summarized as the mean ± standard deviation (SD).

**Statistics**. The results are expressed as a mean ± standard error of the mean of at least three independent experiments. One-way ANOVA followed by Bonferroni's comparison as a post hoc test were used to evaluate the statistical difference of the results between more than two groups. Statistical significance was estimated when $p < 0.05$. Immunoblots, micrographs, and clonogenic assays are representative of at least three biologically independent experiments.

## Data availability

The authors declare that all the data supporting the findings of this study are available within the article and its Supplementary Information files, including the source data file, and from the corresponding author upon reasonable request. The metabolomics data has been deposited in the MetaboLights[34] database with the deposition ID MTBLS2969. Source data are provided with this paper.

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

## Acknowledgements

This work was supported by funds from the following institutions: Agencia Estatal de Investigación/European Regional Development Fund, European Union (PGC2018-096244-B-I00, SAF2016-75442-R), Ministry of Science, Innovation and Universities of Spain, Spanish National Research Council—CSIC, Institut National de la Santé et de la Recherche Médicale —INSERM, Université de Bordeaux, Fondation pour la Recherche Médicale, the Conseil Régional d'Aquitaine, SIRIC-BRIO, Fondation ARC, and Institut Européen de Chimie et Biologie. C.B. was recipient of fellowships from the Minister of Higher Education, Research and Innovation (France) and the Fondation ARC (France). We thank Prof. Patricia Boya (Centro de Investigaciones Biologicas, Madrid, Spain) for kindly providing with the ATG5[+/+] and ATG5[−/−] MEFs. We thank Prof. Benoit Viollet (Institute Cochin, Paris, France) for kindly providing with the AMPK[+/+] and AMPK[−/−] MEFs, and the CA-AMPK plasmid.

## Author contributions

V.H.V. and R.V.D. conceived the project. C.B., V.H.V., P.V., C.F., and R.V.D. designed experiments. C.B., M.T., V.H.V., S.C., A.S.H.C., E.R., M.S., P.V., B.R., C.P.P., and P.D.M. performed experiments. C.B., M.T., A.S.H.C., P.V., C.F., P.D.M., and R.V.D. analysed data. E.B., C.V., P.S., C.F., P.D.M., and R.V.D. secured funding. C.B. and R.V.D. wrote the paper. All authors read and approved the paper.

## Competing interests

The authors declare no competing interests.
