## [Transparent Peer Review File · Nature Communications]

REVIEWER COMMENTS

Reviewer #1 (Remarks to the Author):

The manuscript by the Duran group investigates the cell survival, mTORC1 activation and metabolic adaptations to leucine and glutamine imbalance. They propose that the down-regulation of AMPK and the GABA shunt after glutamine addition may participate in the control of cellular responses. The manuscript contains some potentially interesting observations, which are still disconnected at this stage. The present format of the manuscript is preliminary and the authors do not thoroughly demonstrate their hypothesis.

1) The functional involvement of AMPK is limited by the use of pharmacological activators, some of which have clearly additional targets. They authors claim that their data demonstrate the need of AMPK inhibition by leucine and glutamine in order to regulate mTORC1 and cell survival. However, they never used genetic loss of function approaches. AMPK ko cells from mice or CRISPR-generated should be used to support the role of AMPK.

2) The analysis of autophagy markers may be improved. The LC3 I/II blot in Fig. 1L is not well separated, and fails to show the claimed decrease of LC3 I after metformin/A769662. The images of LQ-treated cells appear at lower magnification as compared to the other conditions (Fig. 1J and 3M).

3) It is not clear how the in vivo evidence of apoptosis after DMKG treatment relates to the rest of the study (Fig. 2J). Is there any evidence for AMPK activation in these cells? DMKG long term treatment and glutaminolysis do not appear to regulate AMPK in vitro (Fig. 3)

4) Similarly, it is not clear whether and how the GABA shunt plays a role in the apoptosis in vitro. Survival responses should be measured after ASNS inhibition alone and in combination to glutaminolysis interference.

Reviewer #2 (Remarks to the Author):

The manuscript “Two metabolic pathways connect glutamine and mTORC1 to control glutamoptosis” by Bodineau et al., investigates glutamine signaling/metabolism in respect to mTORC1 activation. Some of the data was repeating previous work (PMID: 22749528), and there is new data showing glutamine signals to AMPK to regulate mTORC1. Although, the author findings are interesting there are multiple concerns that should be addressed.

Major Concerns:

1) Fig. 1A-B, Sup. Fig. 1A-B. Time courses jump from 2 hrs to a much later time (For example, 24 hrs). Does it take 24 hrs of amino acid starvation to alter ATP/ADP ratio, or can this occur earlier?

2) The authors need to describe in the results and methods sections how these starvation and stimulation (+LQ) experiments were performed in more detail. Why are most of the starvation experiments done for 72 hrs, and when were LQ added back. It appears that LQ were just added to EBSS (starvation media). Did LQ sit in the EBSS media for 72 hours or was it replenished daily. When the samples were harvested 72 hrs later, were there still high free L and Q. Why were the concentrations chosen for L (0.8mM) and Q (2mM). What’s the difference in -AA vs -LQ. With -LQ do you have all the

amino acids but LQ? These details are important in order to interpret the data.

- 3) At what time point does glutaminolysis (Does it start before 24hrs?) start and does that correlate with the ATP/ADP ratio?
- 4) As a control did the authors try other amino acids (+R or +M) to see if they can alter ATP/ADP ratio?
- 5) Does Q or L directly bind to AMPK to regulate its activity. For example, L has been shown to directly bind to and regulate Sestrin2 (PMID: 26449471).
- 6) After 72 hours of starvation I would have expected autophagy to turn on and there to be high mTORC1? For example, after 4 hours of amino acid starvation you can typically see an induction of autophagy. Moreover, a recent paper published that Q alone can regulate autophagy (PMID: 28835610).
- 7) Figure 1J. Is the magnification the same throughout all images? In panel 2, the cells look smaller.
- 8) Figure 1L. There doesn't appear to be changes in the LC3I/II blots. Shouldn't there be 2 bands?
- 9) Figure 3F, should lane 3 be +LQ and Don?
- 10) Figure 3J. Why does +LQ not decrease pAMPK to the same extent as +Q alone. Can the authors include a longer blot of pS6K? It looks like +Q activates mTORC1 to some extent.
- 11) Figure 3P. Does it make a difference if you overexpress RagC or RagD in this experiment? Again +Q looks like it can activate mTORC1 alone.
- 12) Figure 3P. Does +L alone activate mTORC1 with RagB 54L? Some controls are missing, such as +Q and +L with RagB WT and RagB Inactive form.
- 13) Figure 4F. Why is there less +Q in the +AA sample and +QL sample versus +Q alone? Why is there less +L in the +AA sample versus +QL sample and +L alone?
- 14) Figure 4F and sentence starting on page 10 (line 239), "In contrast, the addition of glutamine changed completely the pattern, allowing us to identify a group of metabolites which levels raised very high, particularly GABA, asparagine, glutamate and aspartate". It is true that +Q alone did raise these metabolites, but why didn't the +AA sample and the +QL sample also raise these metabolites, since +Q is present as well?
- 15) Figure 4E: mTORC1 (pS6K and pS6) activity should also be looked at here as well. Figure 4H: pAMPK and pS6K should be assessed here.
- 16) Figure 4H: Dual inhibition of ASNS and GLS decreased mTORC1 activity (~50%), but not completely?
- 17) Can the authors show that AMPK (via the glutamine mechanism) is mediating mTORC1 through its well-known substrates (PRAS40 and TSC)? Or is it through another mechanism?

Minor Concerns:

- 1) The authors should discuss their manuscript in respect to a recent finding where +A regulates AMPK activity (PMID: 30190193).
- 2) For some of the blots pS6K/S6K and pAMPK/AMPK are missing (for example Figure 1L).
- 3) Can the authors clarify in the text why they used U2OS and HCT116 cells for the studies (It is unclear)? Do other cell lines behave the same way?

Reviewer #3 (Remarks to the Author):

Bodineau and colleagues investigate the influence of AMPK and mTORC1 signaling in the mechanisms of apoptosis induced by glutamine in the context of amino acid starvation. This study demonstrates that under extended amino acid starvation, glutamine communicates to mTORC1 signaling through its

consumption by the TCA cycle and the recovery of ATP levels that leads to the inhibition of cell survival signaling pathway; AMP kinase, and the triggering of cell death.

This study is well-executed and addresses a gap in our understanding of the signaling and metabolic mechanisms leading to cell death in response to nutritional stress.

General comments:

One primary concern is the timing of amino acid starvation used in this study. Generally, the authors used a prolonged time (up to 72h starvation) of amino acid starvation to measure cell death, metabolic, and signaling factors (AMPK reactivation, ATP, OCR, etc.). Prolonged amino acid starvation usually induces unfolded protein stress response and probably cell death. The authors should address whether GNC2/ATF4 or other factors play a role in response to LQ stimulation.

The authors are interested in studying cell death or cell survival processes, however, they are parameters that can be used to measure early apoptosis when cells are starved of amino acids and treated with LQ. The authors should measure caspase activation when cells are stimulated with LQ (in the absence of amino acids). The authors' previous work (PMID: 28112156) showed that LQ + amino acid starvation increase BAX levels, one might ask whether AICAR (or metformin) can prevent BAX increase or cytochrome c release induced by LQ.

The authors claim that under the absence of amino acid, glutaminolysis can lead to cell death (glutamoptosis) and that when AMPK activators (AICAR, metformin) are added, cell viability is rescued. This effect is probably due to the induction of autophagy, which enables cell survival. The authors should repeat these experiments in ATG5/7 WT and KO cells to demonstrate the role of autophagy in cell survival when AMPK is reactivated. It is also significant to address whether amino acid starvation and concomitant loss of AMPK activity can mimic LQ addition and lead to apoptosis.

The authors show that the addition of glutamine in the absence of the other amino acids leads to an increase in GABA levels. The authors suggest that GABA participates in the mechanism by which mTORC1 activity is maintained when cells are stimulated with LQ in the absence of the other amino acids. What is the contribution of GABA metabolism to mTORC1 signaling? The authors should knock down glutamate decarboxylase or GABA transaminase and measure mTORC1 signaling in the presence or absence of LQ.

To state that asparagine synthase is required for glutamine-mediated full activation of mTORC1 signaling, the authors should only stimulate with glutamine/leucine +/- BPTES or siRNA GLS1/2 in the presence or absence of siRNA against ASNS. Plus, to be consistent with the other blots, the authors should present 4EBP1 and p-S6K/S6K. Note that it is surprising that GLS inhibition (BPTES) showed no effects on mTORC1 signaling in the siRNA control condition (Fig. 4H).

Specific comments:

The authors should explain why the levels of total AMPK are varying considerably in their conditions (1F, 1I, S1C, 2G, 3F...). This makes the data difficult to interpret.

Fig. 1E: Measurement of 4EBP1 total by western blot should show a mobility shift indicative of the status

of 4EBP1 phosphorylation and not a single band. Is it possible that there is an inversion in the blot between p-4EBP1 and 4EBP1 total? The authors should double-check this matter.

Fig. 1G and 3K: Lamp2 staining should be shown instead of CD63. CD63 does not specifically stain lysosomes but the vacuoles in general.

Fig. 1L: The LC3 blot is not convincing; the authors should present LC3B-I and LC3B-II bands. The authors could use Bafilomycin A to measure autophagy flux under these conditions.

Fig. 2E: Quantification should be presented.

Point-by-point responses to reviewers' comments.

Reviewer #1

The manuscript by the Duran group investigates the cell survival, mTORC1 activation and metabolic adaptations to leucine and glutamine imbalance. They propose that the down-regulation of AMPK and the GABA shunt after glutamine addition may participate in the control of cellular responses. The manuscript contains some potentially interesting observations, which are still disconnected at this stage. The present format of the manuscript is preliminary and the authors do not thoroughly demonstrate their hypothesis.

We thank Reviewer #1 for constructive suggestions. We do believe that, in the new version of the manuscript, we addressed all the concerns raised by all three reviewers, and therefore we now thoroughly demonstrated our hypothesis that AMPK and the GABA shunt actively participate in the cellular response to glutamine sufficiency. In the points below we clarify these concerns one by one, including additional experimental evidence supporting our conclusions.

1) The functional involvement of AMPK is limited by the use of pharmacological activators, some of which have clearly additional targets. They authors claim that their data demonstrate the need of AMPK inhibition by leucine and glutamine in order to regulate mTORC1 and cell survival. However, they never used genetic loss of function approaches. AMPK ko cells from mice or CRISPR-generated should be used to support the role of AMPK.

We thank Reviewer #1 for this fairly valid comment. In this new version of the manuscript, we have included genetic evidence supporting our previous observations obtained with pharmacological approaches. However, we disagree with this reviewer in the exact genetic approach that should be followed. This reviewer proposes a loss-of-function (LOF) approach for AMPK to confirm the effects observed using pharmacological activators of AMPK. However, to “demonstrate the need of AMPK inhibition by leucine and glutamine in order to regulate mTORC1 and cell survival”, a gain-of-function (GOF) approach would be required, rather than AMPK KO cells. For this purpose, we have now included experimental evidence using the expression of a constitutively active mutant of AMPK (CA-AMPK). As we show now in **Figure 1K**, the expression of CA-AMPK prevents almost completely (the endogenous wild type is still present, explaining the lack of a complete prevention) the response of mTORC1 to leucine and glutamine. This result confirmed what we observed using pharmacological activators of AMPK: the inhibition of AMPK in response to glutamine and leucine is a necessary step for the subsequent activation of mTORC1.

Still, and to provide with a complete response to what was requested by this reviewer, we also analysed the response of mTORC1 to glutamine and leucine in AMPK KO MEFs. As we believe that this experimental approach does not address the necessity of AMPK inhibition for mTORC1 activation, we rather opt for not including this results in the manuscript, and we show it here only for Reviewer #1. As can be seen in the **Figure R1** below, the absence of AMPK prevented the activation of mTORC1 in response to glutamine and leucine. This result together with the previous one indicated that, in addition to its inhibition, the presence of AMPK itself is also necessary for the response of mTORC1 to glutamine and leucine. The mechanisms underlying this second effect exceed the scope of this manuscript.

Figure R1. Lack of AMPK prevents the activation of mTORC1 by glutamine and leucine.

Still, we used these AMPK^{-/-} MEFs, to check apoptosis induction under amino acid withdrawal to see whether it affected apoptosis induction. As expected, amino acid withdrawal did not activate mTOR (see new **Supplementary Figure S2A**). As a consequence, amino acid withdrawal did not induce the activation of apoptosis, as determined by the lack of cleaved caspase 3. Indeed, amino acid withdrawal seemed to have a rather protective role in these AMPK^{-/-} MEFs. Thus, we confirmed that mTORC1 activation is necessary for glutamoptosis activation, regardless of the presence of AMPK.

Altogether, both the genetic and the pharmacological approaches demonstrated the need of AMPK inhibition by leucine and glutamine in order to regulate mTORC1. Both AMPK-KO MEFs and CA-AMPK plasmid were kindly provided by Prof. Benoit Viollet (Institute Cochin, Paris, France), to whom we express our gratitude.

2) The analysis of autophagy markers may be improved. The LC3 I/II blot in Fig. 1L is not well separated, and fails to show the claimed decrease of LC3 I after metformin/A769662. The images of LQ-treated cells appear at lower magnification as compared to the other conditions (Fig. 1J and 3M).

We apologise for the low quality of these images. We have now replaced them.

- Figure 1L has been corrected and extended. The decrease of LC3I and the subsequent increase of LC3II are now evident in the new **Figure 1L**. Also, we have now extended this panel, including immunoblots against AMPK (total and phosphorylated) and S6K (total and phosphorylated), fully supporting our conclusions.

- Former Figure 1J, now **Figure 1M**, has been replaced, improving the quality of the image. Scaling is the same for all the panels in **Figure 1M** (differences may raise by the differential activation of mTOR, which affects cell size).

- Similarly, former Figure 3M, now **Figure 4G**, has been replaced and improved. Again, scaling is the same for all the panels in **Figure 4G**.

We believe that with these improvements the analysis of autophagy in response to glutamine and leucine is clear, and our conclusions are fully sustained by the evidence.

3) It is not clear how the in vivo evidence of apoptosis after DMKG treatment relates to the rest of the study (Fig. 2J). Is there any evidence for AMPK activation in these cells? DMKG long term treatment and glutaminolysis do not appear to regulate AMPK in vitro (Fig. 3).

This is a very pertinent question. Still, we want to emphasize that we observed that AMPK did not respond to DMKG *in vitro* in amino acid starved cells. Conversely, our DMKG *in*

in vivo analysis has been performed in fed mice. In these circumstances DMKG would have the potential capacity to inhibit AMPK, and therefore, to overactivate mTORC1, leading to glutamoptosis and cell death. To confirm this possibility, we have now included immunohistochemical analysis of phospho-AMPK in the samples that we previously analysed. As can be shown now in **Supplementary Figure 2D-E**, DMKG treatment in fed mice reduced the basal activation of the xenograft tumors, as determined by the reduced phosphorylation of AMPK (T172). This reduction correlated with the previously observed increase in mTORC1 (S6 phosphorylation) and caspase activation (**Figure 2J-L**). Furthermore, we also observed that metformin co-treatment, completely abolished the capacity of DMKG to inhibit AMPK, leading to mTORC1 inhibition (**Supplementary Figure 2D-E and Figure 2J-L**). Therefore, these results further confirm the role of AMPK in mTORC1-mediated glutamoptosis in xenograft tumors *in vivo*. We also observed that rapamycin treatment not only inhibited mTORC1 in DMKG-treated cells (**Figure 2J-M**), but also abolished AMPK activation in non-treated tumors (**Supplementary Figure 2D-E**), which probably reflects an increase in ATP levels upon mTORC1 inhibition in these tumors. This secondary phenotype, in any case, exceeds the scope of this manuscript, and would require further investigation.

4) Similarly, it is not clear whether and how the GABA shunt plays a role in the apoptosis in vitro. Survival responses should be measured after ASNS inhibition alone and in combination to glutaminolysis interference.

We agree with this reviewer that this can be somehow a confusing point that we tried to clarify. Glutamoptosis requires the full activation of glutaminolysis, to allow mTORC1 activation. Under these circumstances, GLS/GDH are active, and ASNS/GABA shunt does not play a major role. Demonstrating this point, and as suggested by this reviewer, we have now investigated the effect of ASNS and GABA shunt inhibition (silencing indeed) on the glutamoptosis phenotype, both alone and in combination with glutaminolysis interference. As we show now in new **Supplementary Figure S5J-K**, the single inhibition of ASNS or GAD did not affect neither the capacity of glutamine and leucine to activate mTORC1 (perhaps ASNS silencing has a partial effect on it) nor to induce apoptotic cell death. Conversely, the inhibition of GLS (using BPTES), both alone and in combination with ASNS/GAD silencing, is sufficient to prevent mTORC1 activation and apoptotic cell death in response to glutamine and leucine. Thus, the GLS/GDH axis is the main metabolic pathway connecting glutamine and mTORC1/glutamoptosis when leucine is present (and GDH is active).

Contrariwise, we previously showed that ASNS plays a major additive role together with GLS during glutamine sufficiency (in the absence of leucine) in the inhibition of AMPK and the sustainment of ATP levels (see **Figure 5D-E**). We now confirmed that, under these circumstances mTORC1 was not affected, as glutamine alone did not activate mTORC1 (new **Supplementary Figure S5I**). Critically, ASNS together with GLS are necessary for the complete inhibition of AMPK and the subsequent reactivation of mTORC1 in response to all amino acid stimulus (see new extended **Figure 5H**). Indeed, in the presence of all amino acids, GLS inhibition was unable to inhibit mTORC1. However, the combined inhibition of GLS (using BPTES) and ASNS (silencing) completely restored AMPK phosphorylation and prevented mTORC1 activation in the presence of all amino acids. This results further confirmed the major role of ASNS in the response of AMPK/mTORC1 to amino acids.

Therefore, all these results lead us to conclude that glutamoptosis relies on GLS/GDH axis, while ASNS and GABA shunt constitute an alternative pathway to connect glutamine (and amino acid) metabolism with mTORC1 through ATP/AMPK pathway.

Reviewer #2

The manuscript “Two metabolic pathways connect glutamine and mTORC1 to control glutamoptosis” by Bodineau et al., investigates glutamine signaling/metabolism in respect to mTORC1 activation. Some of the data was repeating previous work (PMID: 22749528), and there is new data showing glutamine signals to AMPK to regulate mTORC1. Although, the author findings are interesting there are multiple concerns that should be addressed.

We would like to thank Reviewer #2 for constructive criticism and supportive comments.

Major Concerns:

1) Fig. 1A-B, Sup. Fig. 1A-B. Time courses jump from 2 hrs to a much later time (For example, 24 hrs). Does it take 24 hrs of amino acid starvation to alter ATP/ADP ratio, or can this occur earlier?

As suggested by this reviewer, we have now investigated the drop in ATP levels upon amino acid starvation between 2h and 24h of treatment. As we show now in **Figure 1B** and **Supplementary Figure S1B**, the decrease in ATP/ADP ratio occurred gradually during 2 hours to 24 hours of treatment. But full decrease was observed after 24h of amino acid withdrawal.

2) The authors need to describe in the results and methods sections how these starvation and stimulation (+LQ) experiments were performed in more detail. Why are most of the starvation experiments done for 72 hrs, and when were LQ added back. It appears that LQ were just added to EBSS (starvation media). Did LQ sit in the EBSS media for 72 hours or was it replenished daily. When the samples were harvested 72 hrs later, were there still high free L and Q. Why were the concentrations chosen for L (0.8mM) and Q (2mM). What’s the difference in -AA vs -LQ. With -LQ do you have all the amino acids but LQ? These details are important in order to interpret the data.

We apologise for this confusion, and we modified the Methods section to avoid any misunderstanding. Just to clarify the doubts of this reviewer, let us to explain here:

- In these setups, and all over the manuscript, we never performed starvation/re-stimulation cycles. Conversely, glutamine and leucine (LQ) were added to the starvation medium (EBSS with glucose 25mM) during the whole treatment (72 hours, or as specified). Thus, LQ is present from the beginning in EBSS in LQ-treated cells. The same applies to “Q-treated cells” or “AA-treated cells”.
- The timing of 72 hours was chosen according to our previous publication and previous experience (Villar *et al.*, 2017 Nature Comms 8:14124), in which glutamoptosis is better observed at that timing.
- We did not replenish LQ in EBSS for the whole treatment period (72h). To investigate if there were still high free L and Q in the medium when we harvested the samples 72h later, we measured the levels of both Q and L by mass spectrometry. As we show now in new **Supplementary Figure S1H-I**, the levels of free L or Q was sustained above of the 60% of the initial concentration (0.8mM and 2mM respectively), a concentration high enough to

sustain their consumption by the cells. These chosen concentrations were selected according to media (DMEM) formulation, to match with the concentration used in cell culture for these cells.

- “-AA” and “-LQ” is the same condition. Indeed, we tried to avoid the “-LQ” terminology. We only indicated “-” to emphasize that LQ was not added to the starvation medium.

3) At what time point does glutaminolysis (Does it start before 24hrs?) start and does that correlate with the ATP/ADP ratio?

Again, we apologise for the confusion. We did not “started” glutaminolysis, as we did not re-stimulated cells. LQ was present during the whole treatment. We have now clarified this in the Methods section. Still, to further clarify this point, we measured ATP/ADP ratio during the first 24h of treatment in the presence of LQ. Our new results, shown in **Supplementary Figure S1D-E**, showed that while +LQ condition caused a partial drop in ATP/ADP ratio after 8h of treatment, the levels of ATP/ADP ratio were significantly increased with respect to starving condition (-AA, see **Figure 1B** and **Supplementary Figure S1B**).

4) As a control did the authors try other amino acids (+R or +M) to see if they can alter ATP/ADP ratio?

As requested by Reviewer #2, we measured ATP/ADP ratio in the presence of methionine or arginine. As we show now in **Supplementary Figure S1F-G**, neither methionine nor arginine were sufficient to sustain ATP/ADP ratio, which further confirmed the specificity of glutamine metabolism is this phenotype.

5) Does Q or L directly bind to AMPK to regulate its activity. For example, L has been shown to directly bind to and regulate Sestrin2 (PMID: 26449471).

There is no evidence suggesting a direct binding of Q or L to AMPK. Although we cannot discard it, our results indicated that, if such an interaction would happen, it would be rather irrelevant for the observed phenotype. Indeed, our results indicated that while leucine does not play any role in AMPK inhibition (see **Figure 4D**), the inhibition of GLS and ASNS was sufficient to prevent the inhibition of AMPK by Q (see **Figure 5E**), confirming that glutamine metabolism is necessary for the control of AMPK activity. Thus, a potential interaction between Q/L and AMPK would not affect our model.

6) After 72 hours of starvation I would have expected autophagy to turn on and there to be high mTORC1? For example, after 4 hours of amino acid starvation you can typically see an induction of autophagy. Moreover, a recent paper published that Q alone can regulate autophagy (PMID: 28835610).

mTORC1 re-stimulation due to autophagy activation upon amino acid withdrawal has been reported (including the cited reference) at shorter times (max 5h). These cycles stop operating at longer times. At 72 hours, the capacity of autophagy to re-stimulate mTORC1 has disappeared. Otherwise, mTORC1 would be maintained active continuously upon amino acid starvation, losing its physiological role as switcher in response to nutritional inputs. Other than that, we also observed an activation of autophagy upon 4h of starvation (see **Figure R2**, only for Reviewer #2).

Figure R2. AA starvation induces autophagy after 4h.

7) *Figure 1J. Is the magnification the same throughout all images? In panel 2, the cells look smaller.*

We have now replaced these images, and we hope that new **Figure 1M** has a better quality. We confirmed that magnification is the same in all images. Still, the induction of glutamoptosis and the activation of mTORC1 may affect the shape and size of these cells.

8) *Figure 1L. There doesn't appear to be changes in the LC3I/II blots. Shouldn't there be 2 bands?*

We apologise for this image, which quality was below the minimum requested. We have now replaced this image, and we clearly show in new **Figure 1L** two bands for the LC3I/II blots. Again, please, accept our apologies, and thanks for pointing our attention towards this mistake.

9) *Figure 3F, should lane 3 be + LQ and Don?*

Again, we apologise for this typo, which has been corrected in the new version of the manuscript (new **Figure 3F**).

10) *Figure 3J. Why does +LQ not decrease pAMPK to the same extent as +Q alone. Can the authors include a longer blot of pS6K? It looks like +Q activates mTORC1 to some extent.*

This is a very fine observation that we did not pay much attention to avoid distractive messages. But we also notice this difference. Interestingly, if we compare phospho-AMPK levels and ATP/ADP ratio in response to LQ and Q (see new **Figure 4A** and new **Figure 4D**), we can realise that there is a direct correlation between the capacity of LQ and Q to induce ATP/ADP levels and their respective capacities to inhibit AMPK, thus explaining the results. The reason why Q induces a higher increase in ATP/ADP ratio than LQ is, probably, due to the activation of mTORC1 observed upon LQ addition, not seen in Q condition. This activation of mTORC1 in LQ would likely increase the expenditure of ATP due to an increase in anabolism.

Regarding the capacity of Q to activate mTORC1, indeed we observed a consistent, yet very minor phosphorylation of mTORC1 in response to Q. We have now quantified this phosphorylation (see **Figure R3** below, only for Reviewer #2), and we confirmed that it is quantitatively neglectable.

Figure R3. Q alone induced a consistent yet minor phosphorylation of S6K.

11) Figure 3P. Does it make a difference if you overexpress RagC or RagD in this experiment? Again +Q looks like it can activate mTORC1 alone.

As requested by this reviewer, we have now investigated the overexpression of RagD, both wild type and GDP loaded (constitutively active), to investigate the response to glutamine sufficiency. These results are now shown in new **Supplementary Figure S4P**. As previously observed with RagB, overexpression of a constitutively active form of RagD (RagD77L), and despite the technical issue of a low expression efficiency of this mutant, complemented with glutamine sufficiency to sustain mTORC1 activation as compared with the wild type counterpart (which also exerted some effect, which we cannot quantitatively estimate due to the differential expression of both versions). Thus, the results obtained with RagD confirmed what we observed previously with RagB.

12) Figure 3P. Does +L alone activate mTORC1 with RagB 54L? Some controls are missing, such as +Q and +L with RagB WT and RagB Inactive form.

First of all, we apologise because there is a typo in the mutant nomenclature. We used indeed RagB 99L mutant (which corresponds to RagB GTP loaded, constitutively active mutant), and not RagB 54L.

As requested by this reviewer, we investigated if leucine alone has an effect on mTORC1 upon RagB GTP mutant expression. As we show now in new **Supplementary Figure S4Q**, at these timing leucine alone did not cooperate with RagB GTP to induce mTORC1 activity. Controls using RagB WT are also included. Thus, leucine, at long term, is not sufficient to activate mTORC1, nor to cooperate with RagB GTP to activate mTORC1.

13) Figure 4F. Why is there less +Q in the +AA sample and +QL sample versus +Q alone? Why is there less +L in the +AA sample versus +QL sample and +L alone?

We believe that the reviewer refers to Figure 4A (and not 4F, which does not indicate glutamine or leucine levels). In the case of glutamine, +Q condition does not activate GDH activity (which needs the allosteric activator leucine), which is the main pathway to metabolize glutamine. Thus, in +Q condition there is a decrease in glutamine degradation with respect to +AA and +LQ (where GDH is active). This is reflected in this increased concentration of Q in +Q condition.

Similarly, in the case of leucine the difference probably reflects the inactivation of mTORC1 in +L, while mTORC1 is active in +AA and +LQ. Being an essential amino acid, the differential levels between these conditions likely reflect the lack of usage of leucine in +L condition, due to the absence of protein synthesis, and also likely autophagy activation. In

+AA and +LQ, mTORC1 is active, and thus autophagy is inhibited and protein synthesis active, decreasing the levels of free leucine.

14) Figure 4F and sentence starting on page10 (line 239), “In contrast, the addition of glutamine changed completely the pattern, allowing us to identify a group of metabolites which levels raised very high, particularly GABA, asparagine, glutamate and aspartate”. It is true that +Q alone did raise these metabolites, but why didn’t the +AA sample and the +QL sample also raise these metabolites, since +Q is present as well?

This is indeed a key point in our findings, and we regret that our message was not clear enough to reach the reader of the manuscript. There is a significant metabolic difference between the presence of glutamine (+Q) and the activation of glutaminolysis (+LQ or +AA). As this reviewer pointed out, glutamine is present in all the three conditions (+Q, +LQ, and +AA). Still, glutaminolysis (GLS/GDH) is only active in two of these conditions (+LQ and +AA). As a consequence, we found that glutamine follows an alternative pathway to glutaminolysis when GLS/GDH is not operating (+Q), resulting in a completely different metabolomic pattern, involving ASNS, GABA shunt). In other words, is the absence of leucine, and the subsequent inactivation of glutaminolysis, what triggers the activation of the ASNS/GABA shunt branch. This is important from a therapeutic stand point, as for many years strategies targeting glutaminolysis have been proposed for cancer treatment, and failed. In this work we provide with a biochemical explanation, dissecting this alternative pathway that can circumvent (at least partially) the inactivation of glutaminolysis.

15) Figure 4E: mTORC1 (pS6K and pS6) activity should also be looked at here as well. Figure 4H: pAMPK and pS6K should be assessed here.

As requested, these immunoblots have been included in these images (now **Supplementary Figure S5I** and **Figure 5H**).

16) Figure 4H: Dual inhibition of ASNS and GLS decreased mTORC1 activity (~50%), but not completely?

In the new version of the manuscript we have extended this panel, including immunoblots of the direct mTORC1 targets S6K and 4EBP1. Also, we included now AMPK phosphorylation analysis. According to our blot quantifications (not shown in the manuscript), particularly the mTORC1 direct target S6K, the estimation of mTORC1 inhibition upon ASNS and GLS inhibition was in the range 70-90% inhibition. mTORC1 inhibition during amino acid withdrawal gave a basal phosphorylation of these targets of ca. 80-90%. That means that the inhibition of mTORC1 due to dual inhibition of ASNS and GLS was almost complete, very close to what we observe during amino acid deprivation. We present here the quantification of these blots, in Figure R4 only for Reviewer #2. Still, we agree that certain level of mTORC1 activity may remain (15-25%, particularly for 4EBP1), as these experiments were done in the presence of all amino acids, and some of them may very partially scape the inhibition of ASNS and GLS.

Figure R4. Relative protein phosphorylation of the mTORC1 targets S6K, 4EBP1 and S6 upon dual inhibition of ASNS and GLS.

17) Can the authors show that AMPK (via the glutamine mechanism) is mediating mTORC1 through its well-known substrates (PRAS40 and TSC)? Or is it through another mechanism?

This is a very pertinent point, although somehow exceeds the scope of this manuscript. Indeed, in this manuscript we are not trying to dissect the biochemical connection between AMPK and mTORC1, previously reported and dissected with some detail. Still, we checked one of the direct AMPK targets that has been reported to control mTORC1 activity: RAPTOR phosphorylation at Ser792. This is a well-documented target of AMPK that inhibits mTORC1 activity. As we now show in **Figure 1F** and **Figure 2G** there is a slight yet reproducible decrease in RAPTOR phosphorylation at Ser792 in response to glutamine and leucine treatment, while AICAR treatment reactivated Raptor phosphorylation. These results not only confirmed the inhibition of AMPK activity, but further provided with a direct link between AMPK inhibition and mTORC1 activation.

Minor Concerns:

1) The authors should discuss their manuscript in respect to a recent finding where +A regulates AMPK activity (PMID: 30190193).

Exceptionally, and unfortunately, we disagree here with Reviewer #2. The work indicated by Reviewer #2 [Adachi *et al.*, l-Alanine activates hepatic AMP-activated protein kinase and modulates systemic glucose metabolism (2018) *Molecular Metabolism* 17:61-70, PMID: 30190193], while very interesting, does not relate to the experiments and conclusions that we present in the present manuscript. In that article, the authors elegantly demonstrate that the addition of alanine induces an increase in AMPK activity, and a subsequent decrease in mTORC1 activity specifically in liver derived cells (HepG2, H4IIE, Hepa1, and mouse primary hepatocytes). Although the authors of that article also showed that alanine supplementation cause some changes in TCA intermediates, alanine metabolism does not interferes directly with glutaminolysis, other than affecting transamination. Our metabolomic analysis indicated that alanine levels are increased upon glutamine addition, suggesting that alanine is being produced, rather than consumed, in our setup. In view of these results, we have neither evidence nor arguments to propose that alanine is playing any role during glutamoptosis or even during glutamine sufficiency.

2) For some of the blots pS6K/S6K and pAMPK/AMPK are missing (for example Figure 1L).

We thank Reviewer #2 for pointing our attention towards these flaws in our results. We have now extended these panels (including **Figure 1L**). We hope that now all our immunoblots are as complete as requested by the highest standards of quality.

3) Can the authors clarify in the text why they used U2OS and HCT116 cells for the studies (It is unclear)? Do other cell lines behave the same way?

U2OS and HCT116 cells are cellular models very well characterized and widely used in our laboratory and in the mTOR field. The cellular characterization of glutamoptosis has been previously developed in these cells, together with additional cell lines such as HeLa, Jurkat, HEK293, A549, MEFs, among others. In this manuscript, we have used U2OS and HCT116 to confirm most of the phenotypes. HCT116 were used also for xenograft transplantations. In addition, some phenotypes were confirmed in HEK293 cells and MEFs. We believe this constitutes a good representation of cellular models to confirm our observations.

Reviewer #3 (Remarks to the Author):

Bodineau and colleagues investigate the influence of AMPK and mTORC1 signaling in the mechanisms of apoptosis induced by glutamine in the context of amino acid starvation. This study demonstrates that under extended amino acid starvation, glutamine communicates to mTORC1 signaling through its consumption by the TCA cycle and the recovery of ATP levels that leads to the inhibition of cell survival signaling pathway; AMP kinase, and the triggering of cell death.

This study is well-executed and addresses a gap in our understanding of the signaling and metabolic mechanisms leading to cell death in response to nutritional stress.

We truly thank Reviewer #3 for supportive comments and constructive criticism.

General comments:

One primary concern is the timing of amino acid starvation used in this study. Generally, the authors used a prolonged time (up to 72h starvation) of amino acid starvation to measure cell death, metabolic, and signaling factors (AMPK reactivation, ATP, OCR, etc.). Prolonged amino acid starvation usually induces unfolded protein stress response and probably cell death. The authors should address whether GNC2/ATF4 or other factors play a role in response to LQ stimulation.

This is a very valid comment, but we previously addressed this question in a previous manuscript. In our previous publication Villar *et al.*, 2017 Nature Communications 8:14124, in Supplementary Figure 3I we already showed that UPR does not play a major role in glutamoptosis, using the same setting than we use now in this manuscript. In this figure, several UPR markers were addressed, with negative results. Thus, we concluded that UPR response do not participate in glutamoptosis. We reproduce here, only for Reviewer #3, this panel from that publication.

Figure R5. Neither amino acid starvation nor LQ supplementation at 72h induce UPR. Image taken from Villar *et al.*, 2017 Nature Communications 8:14124, Supplementary Figure S3I.

The authors are interested in studying cell death or cell survival processes, however, they are parameters that can be used to measure early apoptosis when cells are starved of amino acids and treated with LQ. The authors should measure caspase activation when cells are stimulated with LQ (in the absence of amino acids). The authors' previous work (PMID: 28112156) showed that LQ + amino acid starvation increase BAX levels, one might ask whether AICAR (or metformin) can prevent BAX increase or cytochrome c release induced by LQ.

We are a bit confused about this comment by this reviewer. Early apoptosis markers, such as cleaved caspase 3 and cleaved PARP, were already shown in the first submission of this manuscript, together with annexin V/PI staining (see **Figure 2E-G**). Additional apoptotic markers were included in our previous publication (Villar *et al.*, 2017 Nature Communications, PMID: 28112156), including BAX, cleaved caspase 8 and 9, among others. Now, as suggested by this reviewer we have in **Figure 2G** the immunoblot against BAX, which confirmed the induction of apoptosis upon LQ addition. Furthermore, the results in **Figure 2G** indicated that AICAR prevented the upregulation of BAX levels during LQ treatment, further sustaining our conclusions that AMPK inhibition is necessary for glutamoptosis.

The authors claim that under the absence of amino acid, glutaminolysis can lead to cell death (glutamoptosis) and that when AMPK activators (AICAR, metformin) are added, cell viability is rescued. This effect is probably due to the induction of autophagy, which enables cell survival. The authors should repeat these experiments in ATG5/7 WT and KO cells to demonstrate the role of autophagy in cell survival when AMPK is reactivated.

As requested by Reviewer #3, we have analysed ATG5 KO cells to further confirm the role of cell survival when AMPK is reactivated. For this setup, we used ATG5^{+/+} and ATG5^{-/-} MEFs, kindly provided by Prof. Patricia Boya (Centro de Investigaciones Biológicas, Madrid, Spain). As we show now in **Figure 2H-I**, absence of autophagy in ATG5^{-/-} MEFs completely abolished the protective role of AICAR, thus confirming that autophagy activation is necessary for cell survival when AMPK is reactivated.

It is also significant to address whether amino acid starvation and concomitant loss of AMPK activity can mimic LQ addition and lead to apoptosis.

We believe that there is a conceptual mistake in the proposal made by this reviewer. Amino acid starvation and concomitant loss of AMPK will not activate mTORC1, and therefore glutamoptosis will not be activated. Thus, even if AMPK is absent, mTORC1 requires the presence of amino acids to be active. To address this point, we used AMPK^{-/-} MEFs, kindly provided by Prof. Benoit Viollet (Institute Cochin, Paris, France) and we incubated them both in the presence and the absence of amino acids. As expected, amino acid withdrawal did not activate mTOR (see new **Supplementary Figure S2A**). As a consequence, amino acid withdrawal did not induce the activation of apoptosis, as determined by the lack of cleaved caspase 3. Indeed, amino acid withdrawal seemed to have a rather protective role in these AMPK^{-/-} MEFs. Thus, we confirmed that mTORC1 activation is necessary for glutamoptosis activation, regardless of the presence of AMPK.

The authors show that the addition of glutamine in the absence of the other amino acids leads to an increase in GABA levels. The authors suggest that GABA participates in the mechanism by which mTORC1 activity is maintained when cells are stimulated with LQ in the absence of the other amino acids. What is the contribution of GABA metabolism to mTORC1 signaling? The authors should knock down glutamate decarboxylase or GABA transaminase and measure mTORC1 signaling in the presence or absence of LQ.

As requested, we have now investigated the response of mTORC1 and cell death to LQ treatment upon GAD downregulation (silencing). The obtained results are now shown in **Supplementary Figure S5J-K**. These results indicated that GAD downregulation is not sufficient to prevent mTORC1 activation by LQ. This is not surprising, as LQ stimulates glutaminolysis, a condition in which GAD is dispensable. Indeed, in our conclusion, we indicated that GAD plays a role specifically when glutaminolysis is not active, as an alternative pathway. Indeed, the concomitant inhibition of glutaminolysis and GAD prevented mTORC1 activation. However, that impact was similar to the effect observed upon glutaminolysis inhibition, confirming that GAD and GABA shunt pathway are dispensable when glutaminolysis is active. We tried to make this point very clear in our conclusions.

To state that asparagine synthase is required for glutamine-mediated full activation of mTORC1 signaling, the authors should only stimulate with glutamine/leucine +/- BPTES or siRNA GLS1/2 in the presence or absence of siRNA against ASNS. Plus, to be consistent with the other blots, the authors should present 4EBP1 and p-S6K/S6K.

There is a similar argument than in the previous point. As requested by this reviewer, we have investigated the response of mTORC1 cell death to LQ treatment upon ASNS downregulation (silencing). As we show now in **Supplementary Figure S5J-K**, the obtained results indicated that ASNS downregulation is not sufficient to prevent mTORC1 activation by LQ. Again, and as for GAD, this is not surprising, as LQ stimulates glutaminolysis, and therefore the role of ASNS is dispensable. As for GAD, we indicated that ASNS plays a role specifically when glutaminolysis is not active, as an alternative pathway. The inhibition of both glutaminolysis and ASNS also prevented mTORC1 activation, but that impact was similar to the effect observed upon glutaminolysis inhibition, confirming that both ASNS and GAD are dispensable when glutaminolysis is active. We really hope that this point was clear in our conclusions.

Note that it is surprising that GLS inhibition (BPTES) showed no effects on mTORC1 signaling in the siRNA control condition (Fig. 4H).

Former Figure 4H (now **Figure 5H**) indicates the effect of both GLS and ASNS inhibition on mTORC1 activation by all amino acids (not just LQ). In these circumstances, it is not surprising that BPTES did not inhibit mTORC1 pathway (this is indeed known for many years now). The interesting result of this panel is how the concomitant inactivation of both GLS and ASNS almost completely blocked the activation of mTORC1 in response to all amino acids. This result not only confirmed the role of ASNS as an alternative pathway to glutamine metabolism, but also indicated that GLS and ASNS are two necessary elements for the response of mTORC1 to amino acids, further explaining why strategies directed to the inhibition of glutaminolysis failed to stop cancer cell growth.

Specific comments:

The authors should explain why the levels of total AMPK are varying considerably in their conditions (1F, 1I, S1C, 2G, 3F...). This makes the data difficult to interpret.

Slight variations of AMPK levels may reflect some level of degradation during glutamoptosis. Still, we believe that our results clearly allow to detect the variations in AMPK phosphorylation in each condition.

Fig. 1E: Measurement of 4EBP1 total by western blot should show a mobility shift indicative of the status of 4EBP1 phosphorylation and not a single band. Is it possible that there is an inversion in the blot between p-4EBP1 and 4EBP1 total? The authors should double-check this matter.

We double-checked and there is no inversion of the blots.

Fig. 1G and 3K: Lamp2 staining should be shown instead of CD63. CD63 does not specifically stain lysosomes but the vacuoles in general.

As requested by Reviewer #3, we have now included immunofluorescence images using LAMP2 as a lysosomal marker. The results, shown in **Figure 1H** and **Figure 4E**, confirmed what we previously observed using CD63, a marker for late endosomes and lysosomes which we have previously used to trace lysosomal translocation of mTORC1 (see Villar et al., 2017 Nature Comms; Durán et al., 2012 Mol Cell).

Fig. 1L: The LC3 blot is not convincing; the authors should present LC3B-I and LC3B-II bands. The authors could use Bafilomycin A to measure autophagy flux under these conditions.

We apologise for the low quality of these images. As requested, we have now replaced this panel, and LC3 I and II bands can be detected (see new **Figure 1L**). We have also included bafilomycin A treatment in these conditions. As we show in new **Figure 1O**, our results confirmed the decrease in autophagic flux under LQ treatment, and its reactivation upon AICAR addition.

Fig. 2E: Quantification should be presented.

The quantification of the subpopulations of this replicate have been added to the panel (now **Figure 2F**). For the average quantification of late apoptosis of three independent experiments, please refer to new Figure 2E).

REVIEWERS' COMMENTS

Reviewer #1 (Remarks to the Author):

The authors addressed most of the concerns. I still believe the AMPK loss of function experiment is compelling and should be shown. Figure R1 should be made available to the readers as well. It addresses whether AMPK is involved in the suppression of mTORC1 activity after long term Aa. It appears that AMPK ko cells are slightly resistant to mTORC1 suppression after starvation. Surprisingly, AMPK deletion blunts stimulation of mTORC1 by LQ, instead of boosting mTORC1 activity. The authors should discuss the complexity of the metabolic adaptations after these extreme treatments.

Reviewer #2 (Remarks to the Author):

The authors addressed the majority of the concerns.

Reviewer #3 (Remarks to the Author):

The authors have addressed most of my concerns. However, this is unclear to me how asparagine synthase supports ATP synthesis through the TCA cycle in the absence of glutaminolysis. Page 14, line 373-376, the authors state: "Now, our results supported that ASNS metabolizes glutamine, and in combination with GOT1 and GABA shunt, provides with an anaplerotic entry at the TCA cycle for the production of oxaloacetate and the subsequent ATP synthesis in the absence of glutaminolysis." This is a bit convoluted as ASNS converts aspartate and glutamine into asparagine and glutamate, but why is GOT1 mentioned here? Are levels of cytosolic oxaloacetate affected when glutaminolysis is inhibited? The authors do not present any data supporting the role of GOT1 in this process. One experiment to address the direct role of asparagine in controlling mTORC1 in the absence of glutaminolysis is to treat cells with L-asparaginase to deplete asparagine without perturbing ASNS level.

Point-by-point reply to the reviewers' comments

Reviewer #1 (Remarks to the Author):

The authors addressed most of the concerns. I still believe the AMPK loss of function experiment is compelling and should be shown. Figure R1 should be made available to the readers as well. It addresses whether AMPK is involved in the suppression of mTORC1 activity after long term Aa. It appears that AMPK ko cells are slightly resistant to mTORC1 suppression after starvation. Surprisingly, AMPK deletion blunts stimulation of mTORC1 by LQ, instead of boosting mTORC1 activity. The authors should discuss the complexity of the metabolic adaptations after these extreme treatments.

We thank Ref#1 for all the supporting comments and constructive suggestions.

We disagree with the relevance of including this panel in our manuscript. We do agree that the observation that AMPK deletion blunts stimulation of mTORC1 by LQ is certainly unexpected, and indeed it deserves deeper investigations, which we are now conducting. But this observation does not add a significant information to our manuscript, neither to our conclusions.

But despite this disagreement, and following the reviewer's suggestion, we are now including this panel in the manuscript as Sup. Fig 1M, adding the following comment in the manuscript:

"Surprisingly, the ablation of AMPK using AMPK^{-/-} MEFs (again, kindly provided by Prof. Benoit Viollet) also prevented the LQ-mediated activation of mTORC1 (Sup. 1M). This result suggested that, in addition to acting as a negative regulator of mTORC1, the presence of AMPK is necessary for the connection between glutamine metabolism and mTORC1, underscoring the complexity of the metabolic adaptations in these circumstances."

We hope that with this, we comply with requirements of the Ref#1.

Reviewer #2 (Remarks to the Author):

The authors addressed the majority of the concerns.

We thank Ref#2 for all the supporting comments and constructive suggestions.

Reviewer #3 (Remarks to the Author):

The authors have addressed most of my concerns. However, this is unclear to me how asparagine synthase supports ATP synthesis through the TCA cycle in the absence of glutaminolysis.

Page 14, line 373-376, the authors state: "Now, our results supported that ASNS metabolizes glutamine, and in combination with GOT1 and GABA shunt, provides with an anaplerotic entry at the TCA cycle for the production of oxaloacetate and the subsequent ATP synthesis in the absence of glutaminolysis."

This is a bit convoluted as ASNS converts aspartate and glutamine into asparagine and glutamate, but why is GOT1 mentioned here? Are levels of cytosolic oxaloacetate affected when glutaminolysis is

inhibited? The authors do not present any data supporting the role of GOT1 in this process. One experiment to address the direct role of asparagine in controlling mTORC1 in the absence of glutaminolysis is to treat cells with L-asparaginase to deplete asparagine without perturbing ASNS level.

We thank Ref#3 for all the supporting comments and constructive suggestions.

As we tried to summarize in Fig. Sup 5G, the combined action of ASNS, GABA shunt and GOT1 allows the feeding of the TCA cycle in the absence of glutaminolysis, mostly through succinate production (although aKG is also produced, it is only converted from OAA, which therefore does not constitute a net feeding of the cycle, but just a recirculation). The role of GOT1 is necessary for the (partial) reconversion of glutamate into aspartate, to keep feeding ASNS. This conversion is demonstrated by our labelled metabolomic analysis, as labelled carbons coming from glutamine ended up at aspartate and asparagine, which can only be explained by the combined action of GOT1 and ASNS.

The proposed experiment using L-asparaginase would not solve the questions proposed by the reviewer, as L-asparaginase also depletes glutamine (the unspecific glutaminase activity of L-asparaginase is well documented since long ago, see for instance Ollenschläge et al., 1988 Eur J Clin Invest 18(5):512-6). Glutamine depletion would completely interfere with our experimental approach. We do believe that the LOF approach that we followed silencing ASNS (the full KO was not viable), together with the metabolomics analysis, is the best approach to demonstrate the participation of ASNS in the connection between glutamine metabolism and mTORC1.

We hope that, with this, we fully clarified the concerns of Ref#3.